# GenICF: Benchmarking Generative Methods for Inverse Modeling in Inertial Confinement Fusion

## Abstract

The realization of practical inertial fusion energy critically depends on the ability to design laser pulse shapes (LPs) that robustly drive implosions while satisfying stringent physical constraints. Conventional LP design relies on large-scale radiation-hydrodynamic simulations coupled with manual iterative refinement, resulting in high computational cost and limited scalability. Recent advances in generative modeling provide an alternative pathway for data-driven inverse design. In this work, we present the first systematic comparison of generative paradigms for LP design. To enforce physical plausibility, we introduce domain-specific loss formulations. Our results constitute the first principled comparison of generative methods for LP design in inertial confinement fusion (ICF), providing guidance for the development of physics-constrained design frameworks for fusion energy applications.

## 1 Introduction

Inertial Confinement Fusion (ICF) holds exceptional promise as a pathway to clean and sustainable energy, but realizing this potential requires precise control over laser–plasma interactions. A central element of ICF is the laser pulse shape (LP), the temporal profile of laser energy delivery that compresses the fuel pellet and initiates fusion. Designing effective LPs is profoundly challenging: the mapping from laser energy deposition to implosion dynamics is highly nonlinear, multi-scale, and sensitive to small perturbations. Consequently, LP design has traditionally relied on large-scale radiation-hydrodynamic simulations coupled with manual trial-and-error refinement. This process is computationally prohibitive, often requiring weeks of simulation to obtain a single high-performing candidate, and it does not readily generalize across target designs or experimental conditions.

Machine learning offers a compelling alternative by reframing LP design as an inverse generative modeling problem: given desired implosion outcomes, generate candidate pulse shapes that are consistent with both physical constraints and experimental feasibility. However, the comparative strengths and limitations of these approaches for ICF LP design remain unexplored.

In this work, we present the first systematic study of generative models for laser pulse shape design in inertial confinement fusion. Rather than proposing a single architecture, we evaluate multiple generative paradigms and analyze their trade-offs. Specifically, our contributions are:

- We introduce the problem of LP design as a benchmark for evaluating generative models in scientific domains.

- We integrate physics-informed loss formulations to ensure the physical plausibility of generated LPs.

- We provide an assessment of performance in LP design across generative paradigms.

By systematically comparing these approaches, our study establishes a foundation for physics-constrained generative design in ICF.

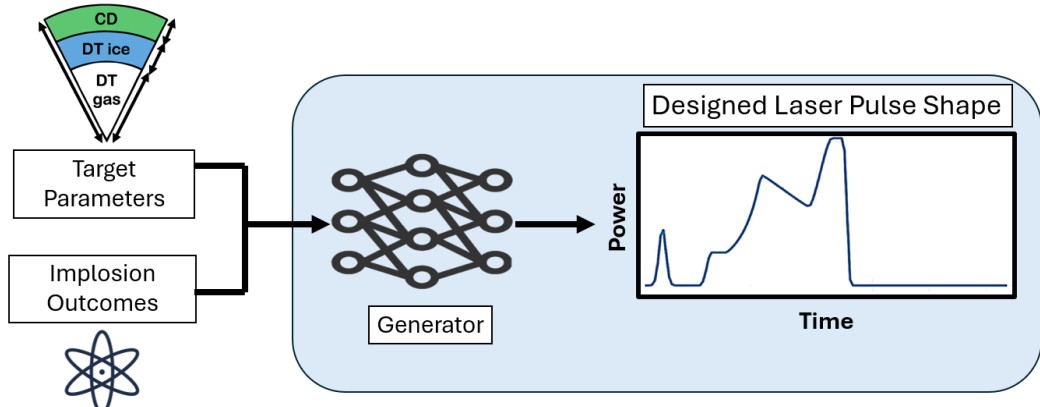

Figure 1: Inverse modeling for LP generation in ICF. Given desired implosion outcomes and target pellet specifications, the model produces a LP that achieves those objectives.

## 2 BACKGROUND

### 2.1 INERTIAL CONFINEMENT FUSION

ICF seeks to achieve nuclear fusion by using high-energy lasers to compress and heat a cryogenic fuel capsule, typically composed of deuterium–tritium, to the extreme densities and temperatures required for thermonuclear ignition (Betti & Hurricane, 2016). A successful implosion must deliver sufficient pressure and temperature while maintaining near-perfect spherical symmetry, a requirement that places stringent demands on the temporal delivery of laser energy.

The LP, the nanosecond-scale temporal profile of laser power, is a primary control knob for implosion dynamics. LPs are typically structured in multiple phases: (i) a low-intensity foot that launches precisely timed shocks, (ii) a rising ramp that regulates pressure buildup, (iii) a high-intensity main drive that accelerates the shell inward, and (iv) in some designs, a trailing tail that sustains compression. Small variations in the relative timing or intensity of these components can dramatically alter implosion performance, making LP design a highly sensitive and complex optimization problem.

Direct experimental exploration of LP design space is impractical due to the cost and scarcity of high-energy laser facilities. As a result, researchers rely heavily on physics-based radiation-hydrodynamic simulations to study implosion dynamics and evaluate candidate LPs. A representative tool is the LILAC code, a one-dimensional Lagrangian hydrodynamics model (Hansen et al., 2018; LLE, 2000; Delettrez et al., 1987; Lees et al., 2021; Gopalaswamy et al., 2019; 2024), which maps LPs and target parameters to implosion outcomes such as neutron yield, shell velocity, and areal density. These simulations provide a controlled and reproducible environment for studying the relationship between LP structure and fusion performance, and they form the basis for evaluating machine learning–based approaches to LP design.

### 2.2 DEEP LEARNING FOR INVERSE MODELING

This subsection provides an overview of the distinct modeling paradigms benchmarked in this work: Auto-regressive Models, Generative Adversarial Networks (GANs), Variational Autoencoders (VAEs), Denoising Diffusion Models, and Flow-based Models. Each model class is included to empirically test how its intrinsic properties—such as sequential generation, latent-space factorization, or iterative denoising—handle the unique nonlinear and multi-scale challenges of Inertial Confinement Fusion (ICF) laser pulse design.

#### 2.2.1 AUTO-REGRESSIVE (AR) MODELS

Auto-regressive (AR) models generate sequences by factorizing the joint probability distribution into a product of conditionals, predicting each element conditioned on all previously generated elements.

Formally, for a sequence $\mathbf{x} = (x_1, x_2, \ldots, x_N)$:

$$p(\mathbf{x}) = \prod_{n=1}^{N} p(x_n \mid x_{1:n-1}), \tag{1}$$

where $x_{1:n-1}$ denotes the subsequence up to position $n-1$.

This factorization allows AR models to capture complex temporal dependencies. While early models used recurrent architectures like LSTMs (Hochreiter & Schmidhuber, 1997), modern AR models are dominated by Transformer architectures (Vaswani et al., 2017), which leverage self-attention to capture long-range dependencies efficiently in parallel.

We explore two main training objectives for AR models: **predictive** and **generative**. Predictive models minimize the mean squared error (MSE) for high reconstruction fidelity:

$$\mathcal{L}_{\text{MSE}}(\theta) = \frac{1}{N} \sum_{n=1}^{N} \| x_n - G_\theta(x_{<n}, \mathbf{c}) \|^2. \tag{2}$$

Generative models, which aim for diversity, minimize the negative log-likelihood (NLL) of the true sequence:

$$\mathcal{L}_{\text{NLL}}(\theta) = -\frac{1}{N} \sum_{n=1}^{N} \log p_\theta(x_n | x_{1:n-1}, \mathbf{c}). \tag{3}$$

We explore various output distributions for $p_\theta$, including Gaussian, Mixture of Gaussians (MoG), and Categorical distribution. AR models are naturally suited for the sequential and temporal nature of laser pulses, offering fine-grained control over individual time steps.

### 2.2.2 GENERATIVE ADVERSARIAL NETWORKS (GANs)

**Generative Adversarial Networks (GANs)** (Goodfellow et al., 2014) learn to synthesize data through an adversarial framework involving a generator $G$ and a discriminator $D$. The generator maps latent variables $\mathbf{z}$ to the data space, while the discriminator distinguishes real samples $\mathbf{x} \sim p_{\text{data}}$ from generated samples $G(\mathbf{z})$. The training is a minimax game:

$$\min_{G} \max_{D}; \mathbb{E}\mathbf{x} \sim p_{\text{data}}[\log D(\mathbf{x})] + \mathbb{E}\mathbf{z} \sim p(\mathbf{z})[\log(1 - D(G(\mathbf{z})))]. \tag{4}$$

GANs are known for generating high-fidelity outputs that closely match the true data distribution. However, training can be characterized by instability, including issues like mode collapse and sensitivity to hyperparameter selection.

For ICF, GANs can provide a powerful mechanism to model the complex, high-fidelity distribution of high-performing laser pulses, allowing us to test their ability to produce physically plausible, constrained designs.

### 2.2.3 VARIATIONAL AUTOENCODERS (VAEs)

**Variational Autoencoders (VAEs)** (Kingma & Welling, 2022) are latent-variable generative models that define a probabilistic mapping between the data space and a compact latent space $\mathbf{z}$. The architecture consists of an encoder $q_\phi(\mathbf{z}|\mathbf{x})$ and a decoder $p_\theta(\mathbf{x}|\mathbf{z})$, trained to maximize the evidence lower bound (ELBO):

$$\mathcal{L}(\theta, \phi; \mathbf{x}) = \mathbb{E}_{\mathbf{z} \sim q\phi(\mathbf{z}|\mathbf{x})}[\log p_\theta(\mathbf{x}|\mathbf{z})] - \text{KL}\big(q_\phi(\mathbf{z}|\mathbf{x}), |, p(\mathbf{z})\big). \tag{5}$$

The KL term enforces that the latent distribution $q_\phi(\mathbf{z}|\mathbf{x})$ remains close to a simple prior $p(\mathbf{z})$, ensuring the latent space is smooth and continuous.

VAEs offer a principled, smooth latent space, which is highly desirable for inverse design problems. This latent manifold can be sampled and interpolated to explore novel regions of the laser pulse design space.

### 2.2.4 DENOISING DIFFUSION MODELS

**Denoising Diffusion Probabilistic Models (DDPMs)** (Ho et al., 2020) are generative models that learn to reverse a gradual forward process that adds noise over $N$ steps. The forward process corrupts input $\mathbf{x}$ into $\mathbf{x}_N$:

$$q(\mathbf{x}_n|\mathbf{x}_{n-1}) = \mathcal{N}\left(\mathbf{x}_n; \sqrt{1 - \beta_n}\,\mathbf{x}_{n-1}, \beta_n\,\mathbf{I}\right). \tag{6}$$

The generative process uses a neural network to approximate the reverse Markov chain, iteratively denoising a Gaussian sample $\mathbf{x}_N$ back to a data sample $\mathbf{x}_0$. This network is trained via a simple noise prediction loss:

$$\mathcal{L}(\theta) = \mathbb{E}_{n,\,\mathbf{x},\,\epsilon}\left[\|\epsilon - \epsilon_\theta(x_n, n)\|^2\right]. \tag{7}$$

The **Denoising Diffusion Implicit Models (DDIM)** (Song et al., 2021) extension allows for a deterministic reverse chain, significantly accelerating sampling time by using fewer, larger time steps.

Diffusion models are state-of-the-art in generating diverse, high-fidelity samples. Their iterative denoising process provides a natural mechanism for controlling the multi-scale and fine-grained features of the laser pulse, making them a strong candidate for this sensitive design problem.

### 2.2.5 FLOW-BASED MODELS

Flow-based generative models construct an invertible mapping ($f$) between a simple base distribution ($\mathbf{z} \sim p_{\mathcal{Z}}$) and the target data distribution ($\mathbf{x} \sim p_{\mathcal{X}}$) using a sequence of bijective transformations (Dinh et al., 2017; Kingma & Dhariwal, 2018). This allows for \*\*exact likelihood computation\*\* via the change-of-variables formula:

$$\log p_{\mathcal{X}}(\mathbf{x}) = \log p_{\mathcal{Z}}(f^{-1}(\mathbf{x})) + \log\left|\det\left(\frac{\partial f^{-1}(\mathbf{x})}{\partial \mathbf{x}}\right)\right|. \tag{8}$$

In this work, we employ the Masked Autoregressive Flow (MAF) (Papamakarios et al., 2017). MAF combines the efficiency of autoregressive transformations with the invertibility required for flows, defining the transformation element-wise:

$$x_n = \mu_\theta(x_{1:n-1}) + \sigma_\theta(x_{1:n-1}) \cdot z_n. \tag{9}$$

Flow-based models are unique in providing tractable density estimation and guarantee perfect invertibility. For the ICF problem, MAF can offer a compelling balance of exact likelihoods for rigorous training and a sequential structure that naturally accommodates temporal signal generation.

## 3 METHODOLOGY

We study generative approaches for designing LPs in ICF by modeling the inverse problem: generating pulse shapes $\mathbf{l}$ that achieve desired implosion outcomes $\mathbf{m}$ given target pellet parameters $\mathbf{p}$. This section describes the general methodology used across all generative models considered in the comparative study.

### 3.1 INVERSE MODELING

Let $\mathbf{m} = \text{LILAC}(\mathbf{l}, \mathbf{p})$ denote the vector of implosion outcomes obtained from the LILAC ICF simulator for a given LP $\mathbf{l}$ and target parameters $\mathbf{p}$. Outcomes include energy yield, areal density, burn width, and ion temperature, while pellet parameters include attributes such as outer radius, ice thickness, and ablator composition.

To train the generative models, we first generate a dataset $D_F = \{(\mathbf{l}_i, \mathbf{p}_i, \mathbf{m}_i)\}_{i=1}^I$ of 1 Million examples via systematic simulation sweeps, capturing the relationship between pulse shapes, target parameters, and resulting implosion outcomes. The LP $\mathbf{l_i}$ is characterized as a real-valued sequence of length 256. $\mathbf{p}_i, \mathbf{m}_i$ are real-valued vectors of size 5 and 12, respectively.

The goal is to learn a mapping $\mathbf{l}' = G_\theta(\mathbf{m}, \mathbf{p})$, where $G_\theta$ is a generative model parameterized by $\theta$, capable of producing diverse LP candidates for given targets. Conditioning inputs $\mathbf{m}$ and $\mathbf{p}$ are embedded via linear transformations before being fed to the model.

## 3.2 GENERAL GENERATIVE FRAMEWORK

All models in the study follow a common generation paradigm:

- **Training:** Models are trained to generate LPs that are consistent with reference data and target outcomes. Depending on the architecture, this may involve sequence prediction, latent denoising, or sampling from a latent distribution.
- **Conditioning:** Models are conditioned on $\mathbf{m}$ and $\mathbf{p}$ to incorporate desired outcomes and pellet properties.
- **Sampling:** New LPs are generated by drawing from the trained model, either sequentially, iteratively, or via latent-space sampling.

This formulation allows evaluation of both reconstruction accuracy and the diversity of generated LPs across different generative paradigms.

### 3.2.1 RECONSTRUCTION OBJECTIVE

A key component of our training objective is the reconstruction loss, which ensures that generated pulses reproduce reference LPs. This objective is a general training signal that encourages the model to generate outputs consistent with the dataset's ground truth.

We define two primary formulations for this loss, depending on the generative model architecture:

- **Pointwise Reconstruction:** For models that produce a deterministic output sequence, the mean squared error (MSE) is used to compare the generated pulse $\mathbf{l}'$ to the reference pulse $\mathbf{l}$.

$$\mathcal{L}(\theta) = \frac{1}{T} \sum_{t=1}^{T} \left| \mathbf{l}_t - \mathbf{l}'_t \right|^2, \quad \mathbf{l}' = G\theta(\mathbf{m}, \mathbf{p}), \tag{10}$$

- **Probabilistic Reconstruction:** For models that output a probability distribution over the pulse values, such as autoregressive models or some diffusion models, the negative log-likelihood of the reference pulse is minimized.

$$\mathcal{L}(\theta) = -\frac{1}{T} \sum_{t=1}^{T} \log p_\theta(\mathbf{l}_t | \mathbf{m}, \mathbf{p}, \text{context}). \tag{11}$$

Here, "context" refers to any additional information available to the model, such as previous time steps ($l_{<t}$) for an autoregressive model, or the noisy input and time step for a diffusion model. This formulation can accommodate various output distributions, including Gaussian or categorical.

This objective provides a strong signal for the generative model, encouraging it to learn the data manifold of plausible LPs.

## 3.3 AUXILIARY OBJECTIVES

While a reconstruction loss enables the model to reproduce existing laser pulses, our broader goal is to facilitate scientific discovery by generating a diverse array of novel, high-performing candidates. To achieve this, we introduce an auxiliary objective that guides the inverse model to prioritize desired implosion outcomes and robustness over exact replication.

A surrogate model $\mathbf{S}_\phi$ approximates the LILAC simulator:

$$\hat{\mathbf{m}} = \mathbf{S}_\phi(\mathbf{l}, \mathbf{p}), \tag{12}$$

where $\phi$ denotes its parameters. The surrogate is trained by minimizing the mean squared error over the simulation dataset $D_F$:

$$\mathcal{L}_{\text{surrogate}}(\phi) = \mathbb{E}_{(\mathbf{l}, \mathbf{m}, \mathbf{p}) \sim D_F} \left[ \| \mathbf{m} - \mathbf{S}_\phi(\mathbf{l}, \mathbf{p}) \|^2 \right]. \tag{13}$$

During training of the generative model $G_\theta$, the surrogate evaluates generated pulses $\mathbf{l}' = G_\theta(\mathbf{m}, \mathbf{p})$, producing an outcome-consistency loss:

$$\mathcal{S}(\theta) = \mathbb{E}_{\mathbf{l}' \sim G_\theta, (\mathbf{m}, \mathbf{p}) \sim D_F} \left[ \|\mathbf{m} - \mathbf{S}_\phi(\mathbf{l}', \mathbf{p})\|^2 \right]. \tag{14}$$

The combined objective is:

$$\mathcal{G}(\theta) = \lambda_\mathcal{L} \mathcal{L}(\theta) + \lambda_\mathcal{S} \mathcal{S}(\theta), \tag{15}$$

where $\lambda_\mathcal{L}$ controls the trade-off between reconstruction fidelity and outcome consistency.

This approach is general and can be applied to all generative architectures considered in the comparative study.

## 3.4 PHYSICS-INFORMED LOSS

To ensure generated LPs respect fundamental physical constraints, we introduce a physics-informed penalty. A primary constraint is energy conservation: the total energy of a generated pulse $\mathbf{l}'$ should not exceed that of a reference pulse $\mathbf{l}$. This is formalized as:

$$\mathcal{P}(\theta) = \mathbb{E}_{\mathbf{l}' \sim G_\theta, \mathbf{l} \sim D_F} \left[ \left( \int_0^T \mathbf{l}'(t) \, dt - \int_0^T \mathbf{l}(t) \, dt \right)^+ \right], \tag{16}$$

where $(\cdot)^+$ denotes the positive part, and $T$ is the total pulse duration.

The final training objective combines reconstruction, outcome-consistency, and physics-informed terms:

$$\mathcal{G}(\theta) = \lambda_\mathcal{L} \mathcal{L}(\theta) + \lambda_\mathcal{S} \mathcal{S}(\theta) + \lambda_\mathcal{P} \mathcal{P}(\theta), \tag{17}$$

where $\lambda_\mathcal{L}, \lambda_\mathcal{S}, \lambda_\mathcal{P}$ control the relative importance of each component. This joint objective is applied uniformly to all models in the study, ensuring a fair comparison of reconstruction fidelity, diversity, outcome consistency, and adherence to physical constraints.

## 4 EXPERIMENTAL EVALUATION

To comprehensively evaluate the performance of various generative modeling paradigms for LP design, we conducted a systematic benchmarking study. The evaluated approaches span a wide range of architectures, including Diffusion Models and Transformers. We also explore several variants of LSTM-based auto-regressive models, namely Deterministic, Gaussian, Mixture of Gaussian, and Categorical. For a broader comparison, we include state-of-the-art generative models from related fields: Tabsyn(Zhang et al., 2024) and Variational Latent Diffusion (VLD) (Shmakov et al., 2023).

All models were evaluated over 5 full training runs and $R = 10$ random seeds on a dedicated test set, $D_{F_{\text{test}}} = \{(\mathbf{l}_e, \mathbf{p}_e, \mathbf{m}_e)\}_{e=1}^E$, which is entirely disjoint from the training data, ensuring a robust assessment of generalization. Our analysis focuses on four critical metrics:

1. **Implosion Outcomes (m) Error:** This metric measures the fidelity of the implosion outcomes produced by the generated LPs. The error is computed via a surrogate loss (Equation 14) and reported as the mean average percentage error, providing a direct measure of design effectiveness.

2. **Reconstruction Error:** This metric quantifies the model's ability to replicate known, high-performing LPs. We calculate the mean squared error (MSE) between the generated pulse $\mathbf{l}'_{e,r}$ and its corresponding ground-truth reference $\mathbf{l}_e$, averaged over all test samples and evaluation runs:

$$\frac{1}{E} \sum_{e=1}^E \frac{1}{R} \sum_{r=1}^R \left\| \mathbf{l}_e - \mathbf{l}'_{e,r} \right\|^2 \tag{18}$$

3. **Generation Diversity:** To assess the models' capacity for scientific exploration, we measure the diversity of generated LPs for a single set of conditioning parameters using two

distinct metrics. The first one is calculated as the average pairwise L2 distance across the generated samples for each test set entry:

$$\frac{1}{E} \sum_{e=1}^{E} \frac{1}{\binom{R}{2}} \sum_{1 \leq j < k \leq R} \left\| \mathbf{l}'_{j,e} - \mathbf{l}'_{k,e} \right\|_2 \tag{19}$$

For context, the estimated upper bound for diversity is 1.9, obtained by comparing randomly selected LPs from the test set.

The second one is calculated as the KNN coverage across the generated samples for each test set sample as:

$$\frac{1}{E} \sum_{e=1}^{E} \frac{1}{R} \sum_{r=1}^{R} d_k(\mathbf{l}'_{e,r}) \tag{20}$$

where $d_k(\mathbf{l}'_{e,r})$ is the Euclidean distance from $\mathbf{l}'_{e,r}$ to its $k$-th nearest neighbor in the set.

4. **Energy Conservation Error:** This is a crucial physics-informed metric that quantifies the percentage deviation of the total energy of the generated pulse from that of the reference pulse. A value closer to zero indicates better adherence to this physical constraint.

$$\frac{1}{E} \sum_{e=1}^{E} \frac{1}{R} \sum_{r=1}^{R} \frac{\left| \int_0^J \mathbf{l}'_{e,r}(j)\, dj - \int_0^J \mathbf{l}_e(j)\, dj \right|}{\int_0^J \mathbf{l}_e(j)\, dj} \times 100 \tag{21}$$

| Approach | L2 ↑ | kNN-Coverage ↑ | m Error ↓ | Reconstruction Error ↓ | Energy Conservation ↓ |
|---|---|---|---|---|---|
| LSTM | – | – | 1.65%±0.09 | 0.0001±5e$^{-5}$ | 0.66%±0.07 |
| Transformer | – | – | 1.94%±0.11 | 0.0008±8e$^{-5}$ | 0.95%±0.1 |
| Diffusion | 0.64±0.077 | 0.37±0.17 | 1.93%±0.046 | 0.0056±2e$^{-4}$ | 1.59%±0.14 |
| LSTM$_{\text{Gaussian}}$ | 0.42 ± 0.02 | 0.31±0.22 | 1.89% ± 0.01 | 0.0005 ± 2e$^{-5}$ | 0.58% ± 0.004 |
| LSTM$_{\text{MixtureOfGaussian}}$ | 0.56±0.1 | 0.36±0.22 | 1.95%±0.09 | 0.0006±8e$^{-5}$ | 1.58%±0.006 |
| LSTM$_{\text{Categorical}}$ | 0.39±0.04 | 0.30±0.22 | 2.01%±0.04 | 0.0009±5e$^{-5}$ | 1.18%±0.08 |
| VAEs | 0.38 ± 0.12 | 0.33±0.18 | 2.75%±0.029 | 0.0021±1e$^{-4}$ | 0.56%±0.12 |
| GANs | 0.60±0.62 | 0.41±0.15 | 5.21%±0.35 | 0.0038±2.5e$^{-4}$ | 0.72%±0.17 |
| Tabsyn | 0.68 ± 0.13 | 0.71±0.072 | 15.8%±0.8 | 0.023±2e$^{-4}$ | 4.76%±0.072 |
| VLD | 0.38 ± 0.1 | 0.25±0.0032 | 16.60%±0.96 | 0.016±6e$^{-5}$ | 1.92%±0.082 |
| FlowMatching | 0.69±0.14 | 0.51±0.11 | 26.40%±2.85 | 0.036±2e$^{-3}$ | 9.14%±0.45 |

Table 1: ICF model performance. $\pm$ denotes standard deviation over seeds. Arrows denote desired improvement direction. For the deterministic models, diversity is not defined.

## 4.1 RESULTS AND DISCUSSION

The experimental results in Table 1 provide a comprehensive view of the performance trade-offs across different generative modeling approaches for LP design.

The deterministic models (LSTM and Transformer) achieve the lowest reconstruction errors and exhibit strong energy conservation, reflecting their high fidelity. However, they produce zero diversity, rendering them unsuitable for exploring the design space.

The probabilistic auto-regressive models offer a more balanced compromise. Among them, the LSTM$_{\text{Gaussian}}$ stands out as the most effective, maintaining low reconstruction error and strong energy conservation while achieving moderate diversity. In contrast, the LSTM$_{\text{MixtureOfGaussian}}$ and LSTM$_{\text{Categorical}}$ achieve greater diversity but at the cost of higher m error and weaker physical consistency.

The Diffusion model also demonstrates notable promise. It produces high diversity, underscoring its ability to generate innovative designs, while keeping m error low. This strength, however, is accompanied by increased reconstruction and energy-conservation errors, highlighting a clear trade-off between diversity and reconstruction fidelity.

By comparison, other generative models, such as VAEs, GANs, and FlowMatching, as well as methods from related domains like Tabsyn and VLD, perform poorly on both m error and reconstruction fidelity.

(a) Fidelity–Diversity trade-off curve.

| $\lambda_{\mathcal{L}}$ | $\lambda_{\mathcal{S}}$ | $\mathcal{L}$ | Diversity |
|------|------|------|------|
| 1.0 | 0.0 | $1.0 \times 10^{-5}$ | 0.130 |
| 0.8 | 0.2 | 0.0031 | 0.21 |
| 0.6 | 0.4 | 0.0039 | 0.37 |
| 0.4 | 0.6 | 0.0056 | 0.50 |
| 0.2 | 0.8 | 0.0064 | 0.66 |

(b) Metrics for different hyperparameters.

Figure 2: Fidelity–Diversity analysis: (a) trade-off curve and (b) measured metrics across $\lambda_{\mathcal{L}}$ and $\lambda_{\mathcal{S}}$ settings.

Our findings identify the autoregressive models and the diffusion model as the most robust and versatile approaches. Both methods strike a meaningful balance between generating novel, diverse designs and adhering to the stringent physical constraints of the problem, offering promising directions for future work in LP design.

## 4.2 ABLATION: TRADE-OFF BETWEEN FIDELITY AND DIVERSITY

We performed an ablation study by systematically varying the relative weights assigned to the reconstruction term ($\lambda_{\mathcal{L}}$) and the outcome-consistency term ($\lambda_{\mathcal{S}}$) in the composite loss 17, while keeping the physics penalty weight ($\lambda_{\mathcal{P}}$) constant. This analysis isolates the inherent trade-off surface between generating highly faithful, but potentially less diverse, laser pulse shapes (high $\lambda_{\mathcal{L}}$) and generating diverse solutions that effectively satisfy the target outcome (high $\lambda_{\mathcal{S}}$).

Figure 2 illustrates this critical trade-off. As the model's objective shifts towards maximizing exploration of the design space (increasing $\lambda_{\mathcal{S}}$), it must tolerate a corresponding degradation in the reconstruction fidelity ($\mathcal{L}$).

The results demonstrate a clear Pareto frontier. When $\lambda_{\mathcal{L}} = 1.0$ and $\lambda_{\mathcal{S}} = 0.0$ (strong emphasis on reconstruction), the model acts primarily as an autoencoder, achieving near-perfect fidelity ($\mathcal{L} = 1.0 \times 10^{-5}$) but minimal design diversity (0.13). Conversely, as the weight shifts to strongly prioritize the surrogate outcome ($\lambda_{\mathcal{S}} = 0.8$), diversity reaches its maximum (0.66) at the expense of fidelity ($\mathcal{L} = 0.0064$). We excluded $\lambda_{\mathcal{L}} = 0.0$ and $\lambda_{\mathcal{S}} = 1.0$ since the model failed in learning physical meaninful representations.

## 4.3 PRACTICAL GUIDANCE FOR INVERSE DESIGN IN ICF

The systematic benchmarking and ablation studies provide concrete, data-driven guidance for selecting the optimal generative strategy based on the specific goals of ICF LP generation. The decision-making process involves two stages: 1) selecting the appropriate generative architecture (Table 1), and 2) tuning the composite loss weights ($\lambda_{\mathcal{L}}, \lambda_{\mathcal{S}}$) to manage the Bias-Variance trade-off (Figure 2).

**Model Architecture Selection** The choice of model is primarily dictated by the required balance between solution fidelity and design diversity(Table 1):

- **Scenario 1: High Fidelity / Low Risk (Interpolation):** If the goal is to generate LPs closely resembling known, successful designs from the training set (e.g., for marginal improvements or verification), models with low Reconstruction Error and low $\mathbf{m}$ Error are preferred. The Deterministic LSTM network is ideal due to its high fidelity and strong Energy Conservation. These models act as robust interpolators but offer no novelty.

- **Scenario 2: Balanced Exploration (Best Generalist):** If the objective is to generate novel, diverse LPs while maintaining strong implosion performance, a probabilistic model is required. The Diffusion Model and LSTM$_{\text{Gaussian}}$ offer the best trade-off. The Diffusion

model offers superior diversity, while the LSTM$_{Gaussian}$ is more physically consistent and computationally cheaper.

**Hyperparameter Guidance**  Once the architecture is chosen, the weights $(\lambda_{\mathcal{L}}, \lambda_{\mathcal{S}})$ must be set based on the desired level of novelty versus realism. The ablation study in Figure 2 explicitly maps this trade-off:

- **Conservative Setting (Minimize Variance):** To minimize the risk of generating non-physical, low-fidelity LPs, one should set $\lambda_{\mathcal{L}} \gg \lambda_{\mathcal{S}}$. This corresponds to the region of the Pareto curve where $\mathcal{L}$ is lowest (e.g., $\lambda_{\mathcal{L}} = 1.0$, $\mathcal{L} = 1.0 \times 10^{-5}$). This strategy minimizes the solution variance but introduces strong bias toward the training distribution, limiting novelty.

- **Exploratory Setting (Minimize Bias):** To maximize the generation of novel LPs (minimizing bias against unexplored regions), one should accept a higher reconstruction loss. This corresponds to the highest diversity region (e.g., $\lambda_{\mathcal{L}} = 0.2$, Div $= 0.8$). This strategy maximizes novelty but increases the variance and risk of generating unstable pulses.

- **Optimal Balance:** For our application, the optimal choice lies at the middle of the Fidelity-Diversity Pareto frontier (e.g., $\lambda_{\mathcal{L}} = 0.4, \lambda_{\mathcal{S}} = 0.6$). This setting maximizes the gain in diversity (novelty) for the smallest corresponding cost in reconstruction fidelity (realism), offering the best overall stability and design utility.

## 4.4 COMPUTATIONAL EFFICIENCY

We analyze the computational overhead for training and inference (sampling) across the different approaches. While the training and inference costs are negligible when considering the month lead time and scale of the ICF experimental cycle, we study these properties because they are paramount for evaluating the model's scalability and applicability to other scientific inverse design tasks where time-to-solution is a critical constraint. All reported times are based on the execution using a single NVIDIA H100 GPU for both training and inference tasks.

Table 2 presents the average training duration, samples generated per second and latency in milliseconds (time per batch). The training time is an average over 5 independent runs, with convergence defined as the point where the validation loss did not meaningfully improve over 20 consecutive epochs. The sampling time reports the average samples/time, when evaluated over the entire test set. These results highlight the intrinsic trade-offs among model complexity, training investment, and broader scientific utility.

| Approach | Avg. Training Time (h) | Throughput (sample/s) | Latency (ms) |
|---|---|---|---|
| LSTM | 10.21±0.23 | 16666 | 26.8 |
| Transformer | 8.42±0.45 | 34666 | 18.3 |
| Diffusion | 29.58±1.4 | 1939 | 528 |
| LSTM$_{Gaussian}$ | 12.21±0.66 | 16666 | 26.8 |
| LSTM$_{MixtureOfGaussian}$ | 13.43±0.75 | 16666 | 26.8 |
| LSTM$_{Categorical}$ | 12.13±0.8 | 16666 | 26.8 |
| VAEs | 32.1±0.37 | 731428 | 14 |
| GANs | 30.46±0.39 | 787692 | 13 |
| Tabsyn | 28.36±1.02 | 293 | 2073 |
| VLD | 24.09±0.58 | 15049 | 68 |
| FlowMatching | 16.23±0.84 | 310303 | 33 |

Table 2: Computational efficiency of the different approaches used for ICF LP generation.

## 5 RELATED WORK

Inverse design, a cornerstone of scientific discovery and engineering, has been revolutionized by deep learning. The goal is to learn an inverse mapping from a desired function or outcome back to

the design parameters that produced it. Our work builds on a growing body of research that leverages generative models for this task. Among the various paradigms, Denoising Diffusion Models have emerged as powerful tools for inverse problems due to their capacity to model complex, high-dimensional data distributions. Their iterative, denoising process allows for generating a diverse set of plausible solutions that are consistent with given constraints, making them ideal for scientific exploration (Daras et al., 2024; Chung et al., 2023; Jalal et al., 2021).

Similarly, autoregressive models have been effectively applied to inverse problems that require capturing long-range dependencies and adhering to intricate constraints, a common need in physics-based applications (Duthé et al., 2025; Luo et al., 2022; Yu et al., 2021). Their sequential nature provides fine-grained control over the generated output, which is particularly relevant for designing temporal signals like a laser pulse. In parallel, GANs have found widespread use in inverse problems, especially in imaging applications (Ray et al., 2021; Anirudh et al., 2018).

VAEs have also been a popular choice for inverse design. By learning a compressed, continuous latent space, VAEs enable a flexible exploration of design parameters and rapid sampling of new designs (Kingma & Welling, 2014; Liu et al., 2021).

Inverse modeling has been successfully applied across a wide range of physical sciences, from estimating material properties in thermodynamics (Ishitsuka & Lin, 2023) to facilitating material discovery (Tamaddon-Jahromi et al., 2020; Haghighat et al., 2021) and enhancing seismic imaging in geophysics (Puzyrev, 2019). Within high-energy and plasma physics, inverse methods have been used for tasks like reconstructing particle dynamics and inferring jet structures in simulations (Shmakov et al., 2023; Ferreira & Carvalho, 2020; Leigh et al., 2024; Öztürk et al., 2024).

However, despite these advances, existing methods are often highly domain-specific and do not directly address the unique challenges of LP design for ICF. The high-dimensional nature of the LP search space, the complex physics-based constraints, and the need for both diversity and precision in the generated pulses present a unique gap in the literature. Our work fills this gap by introducing an study on generative inverse modeling specifically tailored for ICF LP design.

## 6 Conclusion

This study presents the first comprehensive benchmark of generative models for the design of LPs in ICF. Our findings show that some generative paradigms are better suited for this task, which demands a critical balance between design diversity for scientific exploration and strict adherence to physical constraints. The results show that LSTM-based autoregressive models and Diffusion models are the most effective.

The LSTM$_{\text{Gaussian}}$ model stands out for its exceptional fidelity, producing pulse shapes that closely match known, high-performing designs while maintaining excellent physical consistency and moderate diversity. This suggests a powerful capability for generating high-quality, physically plausible designs. In contrast, the Diffusion model excels in its ability to generate a wide range of diverse, novel pulse shapes, making it a powerful tool for exploring new regions of the design space.

Ultimately, this work provides a clear path forward for using generative AI to accelerate the development of inertial fusion energy. The choice of model depends on the specific design goal: a high-fidelity, physically consistent approach for optimizing existing designs, or a more exploratory approach for discovering entirely new solutions. This research lays a crucial foundation for future work, underscoring the potential of tailored generative AI to tackle some of the most complex inverse problems in physics and engineering.

## 7 Reproducibility Statement

The methodology for generating the ICF laser pulse dataset is described in the Appendix. The background on the five distinct generative paradigms (AR, GAN, VAE, Diffusion, and Flow-based models), alongside the full definitions of the novel composite loss function (Equation 17) and the physics-informed penalty, are provided in the main text. Detailed hyperparameter configurations and, libraries used for hyperparameter optimization and data split are provided in the Appendix. The convergence criteria used for all methods is discussed in Section 5. Computing resources and

total computation time for all experiments is discussed in Appendix. Computation time per method is described in Section 4.2. Finally, the analysis of the combined objective's convergence properties and the impact of surrogate error propagation is addressed in the theoretical remarks included in the Appendix.

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

# A APPENDIX

## A.1 EXPERIMENTAL DETAILS

Total computation was conducted using two servers each equipped with four H100 GPUs, running for approximately 600 hours at an average utilization of 75%. During training, we ran various runs in parallel where GPU and CPU capacity allowed. The ICF dataset comprised 1 million simulations, with a 70–30% train–test split used for training and evaluation across all approaches. For training and test set split, we used "train_test_split" function from sklearn.

The hyper-parameters used in our loss function $\mathcal{G}(\theta)$ are shown in Table 8. For the learning rate (LR) we use Pytorch's CosineAnnealing LR scheduler. We used Adam as optimizer. We use $\tau = 0.1$ for the implosion outcome optimization experiments in Section LP Optimization. We used a linear noise scheduler to train the diffusion model. For the forward process we set $N = 100$, while we set $N = 25$ during generation.

The hyperparameters of the models are shown in Table 3 for the denoising network for Diffusion, Table 4 for the LSTM, Table 5 for VAE, Table 6 for GAN, Table 7 for Masked Autoregressive Flow. The hyperparameters were selected using the optuna library (Akiba et al., 2019) using the following ranges: Embedding layers: $[2:4]$, Hidden dimension: $[512:2048]$, Dropout: $[0.1:0.5]$, Learning rate: $[1e^{-6}, 1e^{-2}]$, Batch size: $\{64, 128, 256, 512\}$ and when applicable Transformer layers $[2:4]$, Attention heads: $[4:8]$, Encoder layers $[1:4]$, Decoding layers $[1:4]$.

| Transformer layers | 4 |
|---|---|
| Attention heads | 4 |
| Feed forward dimension | 2048 |
| Embedding layers | 2 |
| Embedding dimension | 1024 |
| Dropout | 0.1 |
| Learning rate | $5e^{-5}$ |
| Batch size | 128 |

Table 3: Network Hyperparameters $\epsilon_\theta$ used in diffusion model.

| Num of layers | 4 |
|---|---|
| Number of hidden units | 512 |
| Learning rate | $1e^{-5}$ |
| Batch size | 128 |

Table 4: Network Hyperparameters used in the LSTM auto-regressive model.

| Encoder layers | 3 |
|---|---|
| Decoder layers | 3 |
| Latent dimension | 64 |
| Hidden dimension (encoder) | 1024 |
| Hidden dimension (decoder) | 1024 |
| Dropout | 0.1 |
| Learning rate | $1e^{-4}$ |
| Batch size | 128 |
| KL divergence weight (beta) | 0.1 |

Table 5: Network Hyperparameters for the VAE model.

The code used for the Tabsyn experiments can be found in `https://github.com/amazon-science/tabsyn` and for VLD in `https://openreview.net/forum?id=v7WWesSiOu&noteId=HimuianB99`, we only employ their end-to-end architecture, but we do not use their consistency loss, since it does not apply to the ICF problem.

| Discriminator steps per iteration | 3 |
|---|---|
| Generator hidden dimension | 1024 |
| Discriminator hidden dimension | 1024 |
| Dropout | 0.1 |
| Generator learning rate | $1e^{-4}$ |
| Discriminator learning rate | $2e^{-4}$ |
| Batch size | 128 |
| Adversarial loss weight | 1.0 |

Table 6: Network Hyperparameters for the GAN model.

| Batch size | 512 |
|---|---|
| Number of coupling layers | 8 |
| Hidden dimension | 512 |
| Layers per coupling | 3 |
| Dropout | 0.1 |
| Learning rate | $1e^{-4}$ |

Table 7: Network Hyperparameters for the Flow model.

| $\lambda_{\mathcal{S}}$ | 0.5 |
|---|---|
| $\lambda_{\mathcal{P}}$ | 0.25 |
| $\lambda_{\mathcal{L}}$ | 0.25 |

Table 8: Hyperparameters used for our inverse model loss function $\mathcal{G}(\theta)$.

## A.2 LILAC SURROGATE

The LILAC surrogate ($\mathbf{S}_\phi$) is a MLP with the following hyper-params. It is trained with the Adam optimizer Kingma & Ba (2014), with a batch size of 128, learning rate of $1e^{-5}$ for 100 epochs.

| Num. of layers | 4 |
|---|---|
| Num. of hidden units | 256 |
| Activation function | ReLU |

Table 9: Hyperparameters used for the LILAC surrogate $\mathbf{S}_\phi$

The surrogate has an average error of 1.4% across all the outputs.

## A.3 MAPPING FUNCTION

The mapping function ($\mathcal{M}$) is a 1 layer bidirectional LSTM with 64 hidden units. We concatenate the hidden state from both directions, before passing it through an output linear layer. It is trained with the Adam optimizer, with a batch size of 64, learning rate of $1e^{-5}$ for 250 epochs. The model achieves an average error of 1.1% across all 12 parameters in $C$.

## A.4 DATA GENERATION

Inputs for experiments and LILAC simulations are specified by target and the laser pulse shape. The laser pulse shape is prescribed using an internal library that parameterizes the laser pulse shape into an ad-hoc descriptive featurization such as the time delay between gaussian shaped features, maximum power and total energy. Therefore, the input is defined within a 20-dimensional parameter space, consisting of target specifications and a parameterized laser-pulse description. Although this space can be uniformly sampled under physical constraints (e.g., maximum laser energy, allowable target thicknesses), random sampling of the 20-D hypervolume does not represent the way experimental and consequently simulation designs are chosen. In practice, high-performing implosions occupy only a small subset of this high-dimensional space, guided by physics intent, rather than being distributed throughout the full admissible hypervolume. Consequently, the region of desirable

inputs does not fill the full 20-D space but instead lies on a lower-dimensional manifold embedded within it. This manifold is not explicitly known or parameterized: it emerges from complex physics couplings between target geometry and laser-pulse features. To more effectively sample this region, we supplement random draws with perturbations of known high-performing implosions ( 400 shots). Perturbing their laser-pulse features and target parameters produces a family of nearby points that remain close to the underlying manifold while still providing variation in implosion performance. This strategy implicitly captures the trait that for a fixed target specification, laser-pulse shapes close to physics-based designs typically outperform large random perturbed ones. The resulting dataset therefore reflects a distribution with multiple high-performance pockets concentrated on an unknown manifold within the 20-D input space, rather than a uniform distribution over the entire hypervolume.

# B  CONSTRAINED OPTIMIZATION

We also try an approach that completely forgoes training a predictive model to estimate the pulse shape by formulating the inverse modeling problem as a constrained optimization problem. The laser pulse $\mathbf{l}$ can be represented as a combination of 12 real-valued parameters $\{c_i\}_{i=1}^N \subseteq \mathbf{c}$ that completely define the shape of the pulse. As described in 3.3, we now train a surrogate $S_\phi$ that maps $\mathbf{c}$ and the target pellet parameters $\mathbf{p}$ to the LILAC outcomes $\mathbf{m}'$. The inverse problem can now be formulated as a constrained optimization problem over the parameters $\{c_i\}_{i=1}^{12}$ that minimizes the following mean squared error.

$$\mathcal{L}_{\text{surrogate}}(\phi) = \mathbb{E}_{(\mathbf{c},\mathbf{m},\mathbf{p})\sim D_F} \left[ \|\mathbf{m} - \mathbf{S}_\phi(\mathbf{c},\mathbf{p})\|^2 \right]. \tag{22}$$

Note that since $S_\phi$ is differentiable, we can also use gradient-based optimization techniques. The pitfall of this approach is that it often produces pulses that are out-of-distribution for the surrogate $S_\phi$ leading to predictions $\mathbf{m}'$ that are unreliable. Also, the optimization routine needs to be run for every sample. With this technique, we get an implosion outcomes error of 8%.

# C  THEORETICAL REMARKS

## C.1  CONVERGENCE OF THE COMPOSITE OBJECTIVE

Equation 17 defines the composite loss

$$\mathcal{G}(\theta) = \lambda_{\mathcal{L}}\mathcal{L}(\theta) + \lambda_{\mathcal{S}}\mathcal{S}(\theta) + \lambda_{\mathcal{P}}\mathcal{P}(\theta), \tag{23}$$

where $\mathcal{L}(\theta)$ denotes the reconstruction or likelihood term, $\mathcal{S}$ the outcome-consistency term using the differentiable surrogate $S_\phi$, and $\mathcal{P}(\theta)$) the physics penalty. Because $\mathcal{G}(\theta)$ is a nonconvex objective, global optimality cannot be guaranteed. However, under standard smoothness and bounded-variance assumptions for stochastic gradients, stochastic gradient descent (SGD) with a suitable step-size schedule converges to a first-order stationary point in expectation. Formally, if $\nabla_\theta\mathcal{G}$ is $L_G$–Lipschitz and stochastic gradients $\hat{g}_t$ satisfy $\mathbb{E}[\|\hat{g}_t - \nabla\mathcal{G}(\theta_t)\|^2] \leq \sigma^2$, then with constant step size $\eta < 1/L_{\mathcal{G}}$ we obtain the standard rate $\mathbb{E}\|\nabla\mathcal{G}(\theta_T)\|^2 \leq O(1/T) + O(\eta\sigma^2)$. In practice, adaptive methods such as Adam used in our experiments empirically achieve similar convergence behavior. This statement clarifies that the optimization of Equation 17 yields approximate stationary points of the joint empirical objective.

## C.2  ANALYSIS OF SURROGATE-ERROR PROPAGATION

Below we provide an analysis of how surrogate approximation error affects the surrogate-based outcome-consistency loss. This makes explicit the mechanisms by which a nonzero surrogate error can propagate through the inverse model.

Let $\mathbf{l}' = G_\theta(\mathbf{p},\mathbf{m})$ be a generated pulse for condition $p$, let

$$r_\phi(\mathbf{l}') := S_\phi(\mathbf{l}',\mathbf{p}) - \mathbf{m}, \qquad r_{\text{true}}(\mathbf{l}') := \text{LILAC}(\mathbf{l}',\mathbf{p}) - \mathbf{m},$$

and define the pointwise surrogate error

$$\varepsilon_\phi(\mathbf{l}') := S_\phi(\mathbf{l}', p) - \text{LILAC}(\mathbf{l}', p) = r_\phi(\mathbf{l}') - r_{\text{true}}(\mathbf{l}').$$

The surrogate-based outcome-consistency term is

$$\mathcal{L}_S = \|r_\phi(\mathbf{l}')\|^2.$$

Noting $r_\phi = r_{\text{true}} + \varepsilon_\phi$ we have the exact identity

$$\mathcal{L}_S = \|r_{\text{true}}(\mathbf{l}') + \varepsilon_\phi(\mathbf{l}')\|^2.$$

Applying the reverse triangle inequality yields the simple bound

$$\left| \sqrt{\mathcal{L}_S} - \|r_{\text{true}}(\mathbf{l}')\| \right| \le \|\varepsilon_\phi(\mathbf{l}')\|.$$

Thus, pointwise surrogate error translates directly into an additive bound on the surrogate-implied outcome error: the surrogate loss is a proxy for the true outcome error up to the pointwise surrogate deviation.

### C.3   SURROGATE LOSS ROBUSTNESS

The primary risk of using a surrogate model ($S_\phi$) in the loss function $\mathcal{S}(\theta)$ is that the generator ($G_\theta$) may exploit the surrogate's approximation error $\varepsilon_\phi$ to satisfy the loss function $\mathcal{S}(\theta)$ without satisfying the true physical objective.

**Conditional Guarantee**: The generator $G_\theta$ minimizes the true LILAC residual, provided that the surrogate's pointwise approximation error, $\varepsilon_\phi(\mathbf{l}') := S_\phi(\mathbf{l}', \mathbf{p}) - \text{LILAC}(\mathbf{l}', \mathbf{p})$, is uniformly small across the entire region of the laser pulse (LP) space explored by the generator.Our analysis of the loss propagation (Section X, Appendix A) shows the exact relationship:

$$\left| \sqrt{\mathcal{S}(\theta)} - \|r_{\text{true}}(\mathbf{l}')\| \right| \le \|\varepsilon_\phi(\mathbf{l}')\|$$

where $\sqrt{\mathcal{S}(\theta)}$ is the surrogate-implied outcome error and $\|r_{\text{true}}(\mathbf{l}')\|$ is the true LILAC outcome error. This identity demonstrates that the two objectives are functionally equivalent up to the magnitude of the surrogate error.

**Adversarial Exploitation**: The robustness fails if the generator produces an LP, $\mathbf{l}^*$, that is an adversarial example to the surrogate. This occurs when $\mathbf{l}^*$ lies in a region of the design space where the surrogate $S_\phi$ is highly inaccurate ($\|\varepsilon_\phi(\mathbf{l}^*)\|$ is large). In this scenario, the generator may achieve low error on $\mathcal{S}(\theta)$ while the true LILAC error remains large ($\|r_{\text{true}}(\mathbf{l}^*)\| \gg 0$). This constitutes explatation of the surrogate and generates a physically misleading design. The inclusion of the reconstruction loss ($\mathcal{L}(\theta)$) and physics penalty ($\mathcal{P}(\theta)$) are necessary to mitigate this by filtering physically implausible designs. This is confirmed by our experiments in Section 4.2, where a model without $\mathcal{L}(\theta)$ fails to learn physical viable pulses.

### C.4   IDENTIFIABILITY AND THE ROLE OF $\lambda_\mathcal{L}$

The ICF LP inverse design problem is a classic example of an ill-posed inverse problem, meaning the solution is non-identifiable: multiple distinct LPs ($\mathbf{l}'_1 \ne \mathbf{l}'_2$) can produce close fusion outcomes ($\mathbf{m}$).

**Inverse Problem Constraint**: Minimizing the outcome loss ($\lambda_\mathcal{S} \mathcal{S}(\theta)$) alone would force the generator to converge to any LP that satisfies the desired outcome, likely producing an average or non-physical pulse shape located far from the actual manifold of feasible LPs (as described in Section 4.2).

**Regularization by** $\lambda_\mathcal{L}$: The Reconstruction Loss ($\lambda_\mathcal{L} \mathcal{L}(\theta)$) is mathematically required to regularize the non-identifiable problem. By enforcing fidelity to the training data manifold, $\mathcal{L}(\theta)$ forces the generated solution to be the closest physically plausible pulse to the training set that also satisfies the outcome constraint. $\lambda_\mathcal{L}$ effectively selects a unique, stable solution from the infinite set of LPs that minimize $\mathcal{S}(\theta)$.

C.5 BIAS–VARIANCE TRADE-OFFS

The ablation study varying $(\lambda_{\mathcal{L}}, \lambda_{\mathcal{S}})$ (Figure 2b) explicitly maps the bias–variance trade-off in the generated output:

**Variance ($\lambda_{\mathcal{S}}$ dominance)**: A low $\lambda_{\mathcal{L}}$ encourages the generator to explore further from the training manifold to satisfy the outcome, thus leading to high diversity and an increased variance.

**Bias ($\lambda_{\mathcal{L}}$ dominance)**: A high $\lambda_{\mathcal{L}}$ severely constrains the generator to the known training data, leading to low diversity (low variance). This strategy is robust but prone to bias against novel pulse shapes that may lie slightly outside the training data distribution.

# D CORRELATION BETWEEN OUTCOME-CONSISTENCY ERROR AND OUT-OF-DISTRIBUTION DATA

A key concern when employing surrogate models is whether their approximation error remains stable when the generator produces pulses that differ from those seen during surrogate training. To quantify this effect, we compare the implosion outcomes error for generated pulses with a simple novelty measure computed relative to the original training dataset.

**Novelty score.** Given a generated pulse $\mathbf{l}'$, we define its novelty as its distance to the closest pulse in the training set $\mathcal{D}_{\text{train}}$:

$$d(\mathbf{l}') = \min_{\mathbf{l} \in \mathcal{D}_{\text{train}}} \|\mathbf{l} - \mathbf{l}'\|_2.$$

This quantity measures how far the generator has moved away from the region where the surrogate has been trained and validated, and therefore acts as a proxy for out-of-distribution (OOD) deviation.

**Implosion Outcomes error.** We compute the implosion outcome error

$$e(\mathbf{l}') = \left[ \|\mathbf{m} - \mathbf{S}_{\phi}(\mathbf{l}', \mathbf{p})\|^2 \right].$$

The magnitude $e(\mathbf{l}')$ reflects the error calculated by our surrogate model $\mathbf{S}_{\phi}$ which will be propagated trough our loss function in Equation 14 pulse.

**Correlation analysis.** To assess whether $e(\mathbf{l}')$ systematically increases as pulses move farther from the training manifold, we compute the Spearman rank correlation

$$\rho_{\text{Spearman}} = \text{corr}_{\text{Spearman}}(\{d(x)\}, \{e(x)\}).$$

Spearman correlation is insensitive to scale and captures monotonic relationships, making it appropriate for detecting trends such as "greater novelty $\Rightarrow$ larger surrogate error". A strong positive correlation would indicate that $\mathbf{S}_{\phi}$ introduces a strong bias in the generative model against OOD regions, while a correlation close to zero would suggest that the surrogate maintains relatively uniform influence across the generator's operating domain.

We ran this evaluation over the test set using Diffusion as base model since it offers a good balance between **m** error and **diversity**. The Spearman correlation between **m** error magnitude and the novelty score on the validation set is $\rho_{\text{Spearman}} = \mathbf{0.353}$ with $p < 10^{-290}$. This modest correlation indicates that the surrogate error exhibits a mild monotonic increase with distance from the training manifold. The effect size is not strong, suggesting that the surrogate remains reasonably reliable across the region of pulse space that the generative models operate in. Importantly, this behavior is consistent with expectations for neural surrogates trained on physically coherent waveform datasets: their accuracy degrades smoothly with novelty but does not break down or dramatically inflate in regions relevant to our experiments.

Moreover, as shown by the results obtained by the diffusion model, diversity does not negatively affect **m** since it is able to achieve high diversity while maintaining low **m** error.

# E PULSE SHAPE CONSTRAINED DESIGN

In many ICF experimental design scenarios, scientists require control over specific regions or attributes of the LP. A mapping function $\mathcal{M}$ (details in A.3) is used to project an LP **l** onto $M = 12$

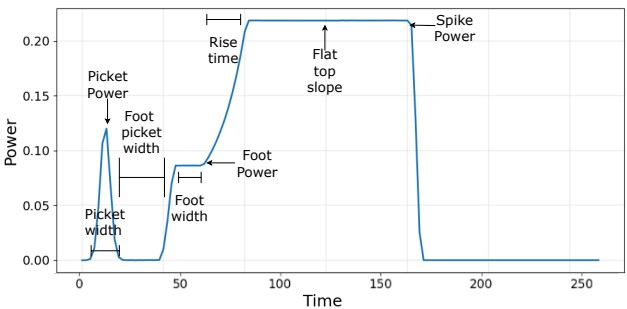

Figure 3: The LP is typically characterized by 12 parameters, some of which are annotated here.

parameters $C = \mathcal{M}(\mathbf{l}) = \{c^1, \ldots, c^{12}\}$. Each $c^m$ corresponds to a physically interpretable property of the LP (Figure 3). When constructing new designs, scientists fix one or two of these values within $C$, while allowing the remaining parameters, and thus the shape of the pulse, to vary.

However, our current inverse model $G_\theta$ lacks a mechanism to enforce such partial constraints during inference. A straightforward solution would be to take the constraint parameter $\mathbf{c}^m$ as an additional input during training, producing an LP conditioned on the desired implosion outcomes $\mathbf{m}$, target parameters $\mathbf{p}$, and the specified constraint as $\mathbf{l}' = G_\theta(\mathbf{m}, \mathbf{p}, \mathbf{c}^m)$. A constraining loss can then be introduced to enforce consistency between the specified constraint values and their corresponding values in the generated LP:

$$\mathcal{J}(\theta) = \mathbb{E}_{\mathbf{l}', c^m} \left[ \| c_{l'}^m - c^m \|^2 \right] \tag{24}$$

where $c_{l'}^m \in C_{l'}, C_{l'} = \mathcal{M}(\mathbf{l}')$. The constrained $G_\theta$ objective can be defined as:

$$\mathcal{G}(\theta) = \lambda_{\mathcal{L}} \mathcal{L}(\theta) + \mathcal{S}(\theta) + \lambda_{\mathcal{P}} \mathcal{P}(\theta) + \lambda_{\mathcal{J}} \mathcal{J}(\theta) \tag{25}$$

### E.1 DIFFUSION-BASED ADAPTATION

The above approach enables $G_\theta$ to respect defined constraints. However, exhaustively training separate models for all combinations of one or two parameters in $C$ is computationally infeasible due to combinatorial complexity.

To address this, inspired by few-shot learning in diffusion and inverse modeling (Bartunov & Vetrov, 2018; Netanyahu et al., 2024; Giannone et al., 2022; Hu et al., 2024), we propose a rapid adaptation strategy. Instead of retraining, we adapt a pre-trained model $G_\theta(\mathbf{m}, \mathbf{p})$ via a few gradient updates to satisfy constraint $c$.

Our adaptation approach employs an embedding network $\mathcal{B}_\delta$ that encodes constraint parameters $c \in C$ to condition LP generation. This results in a conditioned noise model: $\epsilon_\theta(\mathbf{l}_n, n', \mathbf{m}', \mathbf{p}', \mathcal{B}_\delta(c))$. To simplify conditioning, we enforce equal embedding dimensionality for $n'$ and $c' = \mathcal{B}_\delta(c)$, enabling their combination into a single conditioning variable $u' = n' + c'$. This allows us to condition the model as $\epsilon_\theta(\mathbf{l}_t, u', \mathbf{m}', \mathbf{p}')$, requiring no architectural changes to the original model. During adaptation, only the parameters of $\mathcal{B}_\delta$ are fine-tuned, according to Equation 24, while the remaining weights are frozen. This strategy facilitates efficient, stable adaptation, preserving model integrity and accuracy while effectively enforcing constraints. Examples of this adaptation technique for picket power as a constraint are shown in Figure 4. Importantly, for a specific constraint $c$, the model is fine-tuned once, and this fine-tuned model can then be used to generate an arbitrary number of pulses with different specifications $\mathbf{m}$ and $\mathbf{p}$.

### E.2 GRADIENT-BASED ADAPTATION

Alternatively, the LP can also be adapted at post-design using a model-agnostic technique without additional finetuning. To honor constraints in $\mathcal{C}$ while maintaining desired implosion outcomes, a loss combining the pulse constraining loss $\mathcal{J}(\theta)$ (Equation 24) and the LILAC surrogate loss $\mathcal{S}(\theta)$ (Equation 14) is formulated. Since both the losses are differentiable w.r.t the pulse $\mathbf{l}'$ (now referred to as $\mathcal{J}(\mathbf{l}'), \mathcal{S}(\mathbf{l}')$), we can perform gradient descent with respect to the following loss and update

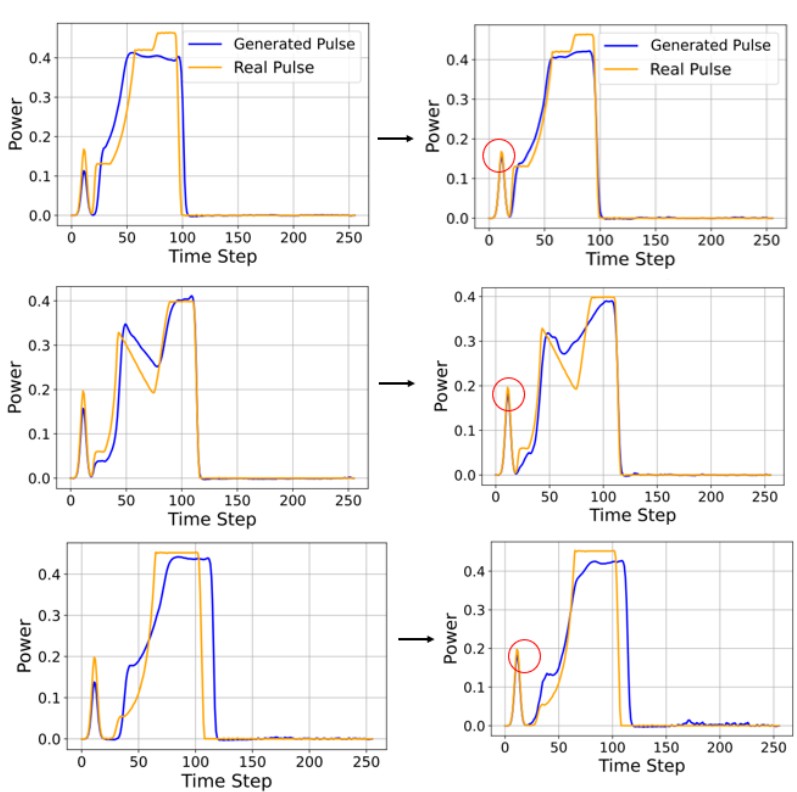

Figure 4: LPs constraining on Picket Power (red highlight). The model successfully constrains the picket power to the target value while allowing variation in the rest of the LP.

the pulse $l'$:

$$\mathcal{T}(l') = \mathcal{S}(l') + \lambda_{\mathcal{J}}\mathcal{J}(l') \tag{26}$$

$$l' := l' - \alpha\nabla_{l'}\mathcal{T}(l') \tag{27}$$

where $\alpha$ is the learning rate. A disadvantage of this technique is that gradient descent can lead the LP into undesirable territory, such as negative values or peaks that violate the maximum power. However, in practice, we find that these issues can be effectively mitigated by clipping the pulse. Moreover, gradient-based adaptation must be performed for each unique constrained LP, whereas our Diffusion-Based Adaptation allows a fine-tuned model to be reused indefinitely.

### E.3 LP CONSTRAINING PERFORMANCE

We evaluate the ability of our framework to rapidly adapt to specific design constraints $c$, using both Diffusion-Based and Gradient-Based adaptation strategies. For the Diffusion-Based approach, the model is fine-tuned for only 10 epochs on a new constraint parameter $c^m$, requiring minimal computational overhead. In parallel, we apply the Gradient-Based adaptation to the deterministic models LSTM and Transformer, allowing for direct post-generation refinement.

We focus our evaluation on two physically meaningful parameters: picket power and foot power (see Figure 3). To quantify adaptation accuracy, we compute the mean absolute percentage error (MAPE) between the specified constraint value $c$ and the corresponding value extracted from the generated pulse (as defined in Equation 24). The Gradient-Based approach achieves a MAPE of $\mathbf{2.5\% \pm 0.12}$, while the Diffusion-Based method yields a MAPE of $\mathbf{3.4\% \pm 0.14}$. Importantly, both adaptation methods preserve performance on the target implosion outcomes.

These constraint adaptation strategies empower scientists to generate diverse, high-quality pulse shapes that meet desired implosion outcomes and can be interactively tuned to satisfy evolving experimental or engineering constraints.

### E.4 INPAINTING/PROMPTING

For additional controllability, we support inpainting-based generation, enabling scientists to specify desired LP design directly in the LP space. Rather than conditioning on the $\mathcal{C}$ constraining parameters, users can provide a partial LP such as prefixes or fixed regions. For the auto-regressive models, providing a LP prefix is akin to prompting it to complete the rest of the pulse. For diffusion, the model conditions its denoising process on specified LP segments, producing coherent, physically valid completions. Table 10 shows the results when evaluating our system following a predefined section of the pulse for inpainting. For the auto-regressive models, a prefix comprising 10% of the pulse (this corresponds to the first peak) is provided as the prompt. For diffusion random segments of the LP are specified, and the rest of the LP is masked. Examples of the LPs generated with this technique can be found in Figure 5 and Figure 6.

| Approach | m Error ↓ | Reconstruction Error ↓ |
|----------|-----------|------------------------|
| Diffusion | $2.0\% \pm 0.012$ | $0.002 \pm 2.32e^{-6}$ |
| LSTM | $1.55\%$ | $0.0003$ |

Table 10: Model performance when inpainting for a specific pulse shape region. $\pm$ denotes standard deviation over seeds.

## F FAILURE MODES

### F.1 PHYSICAL VIOLATIONS

This section reports the violation rates for the two primary operational constraints investigated: the instantaneous per-timestep maximum power caps and the maximum cumulative energy allowed during ICF experiments. These rates, detailed in Table 11, are crucial indicators of the robustness and feasibility of the generated LPs.

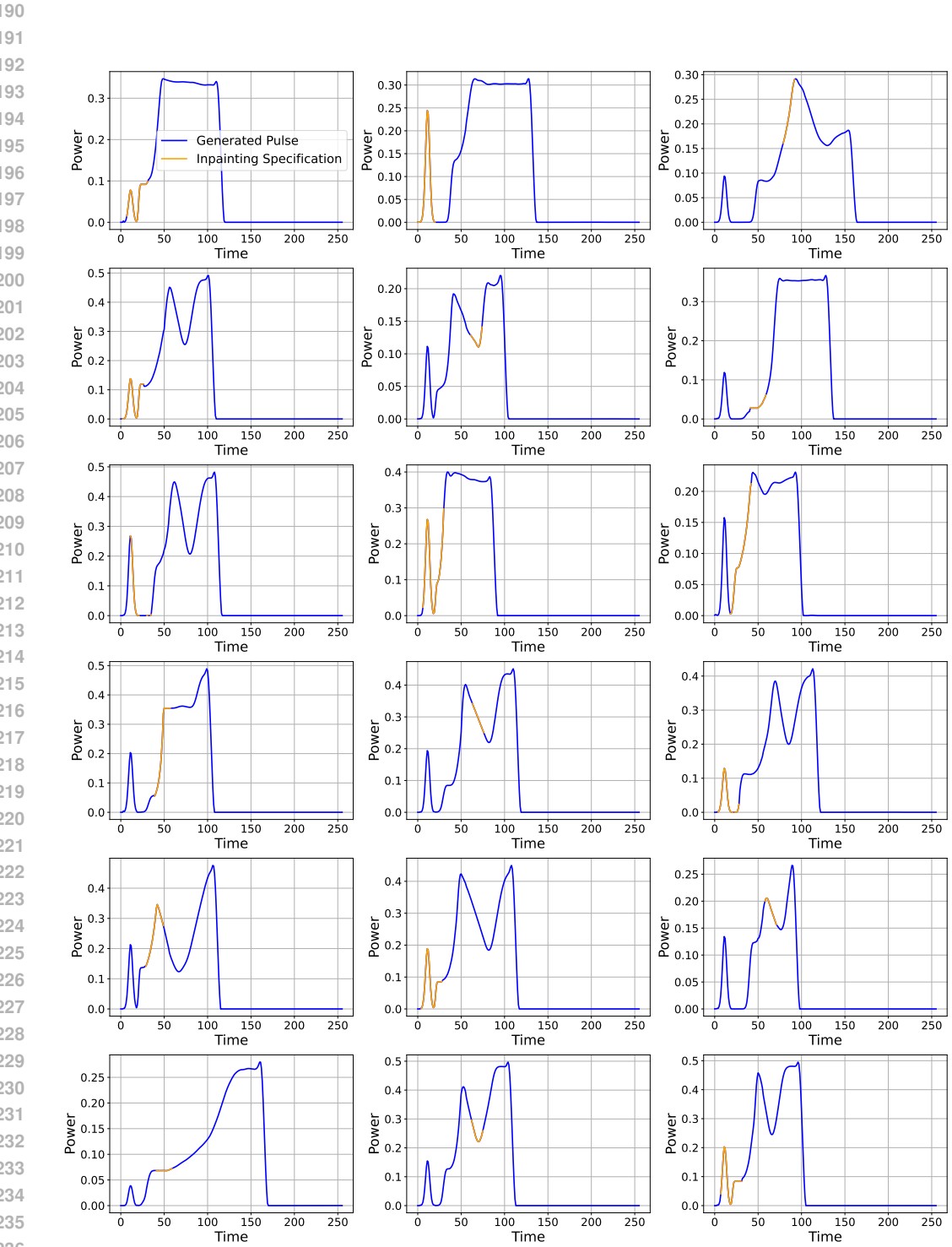

Figure 5: LPs generated by the model when doing inpainting using Diffusion.

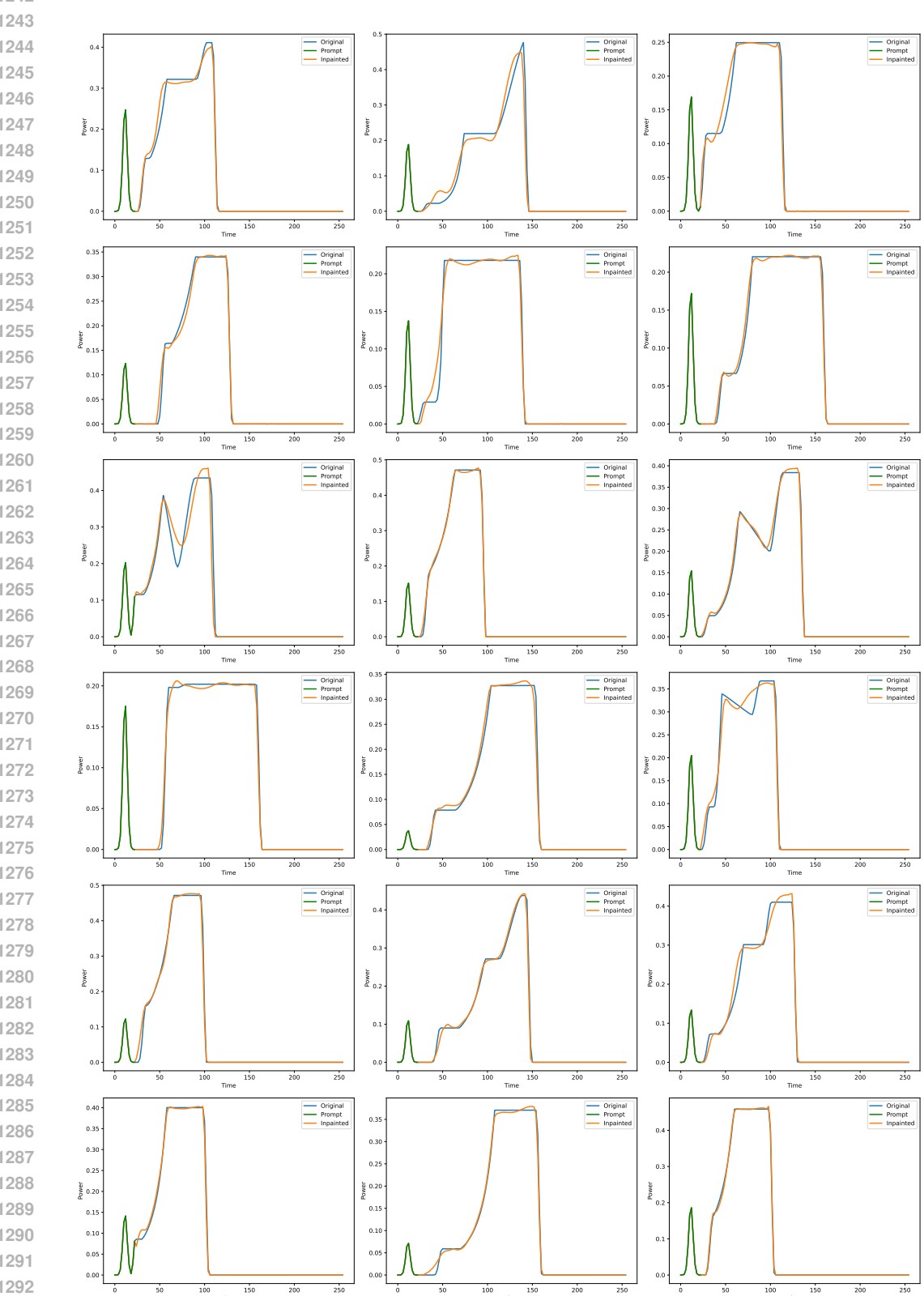

Figure 6: LPs generated by the model when doing inpainting using the LSTM.

| Approach | Maximum Power Cap | Maximum Cumulative Energy |
|---|---|---|
| LSTM | 0% | 0% |
| Transformer | 0% | 0% |
| Diffusion | 0% | 0% |
| $LSTM_{Gaussian}$ | 0% | 0% |
| $LSTM_{MixtureOfGaussian}$ | 0% | 0% |
| $LSTM_{Categorical}$ | 0% | 0% |
| VAEs | 0% | 0% |
| GANs | 0% | 0% |
| Tabsyn | 7.69%±0.74 | 0.26 %± 0.08 |
| VLD | 0% | 0% |
| FlowMatching | 2.45%±0.23 | 0.02%±0.001 |

Table 11: Violation Rates of the bench-marked approaches for LP Generation.

### F.1.1 ANALYSIS OF CONSTRAINT ADHERENCE

The data in Table 11 reveals a clear division in constraint adherence among the tested generative models. A substantial majority of approaches, including all LSTM variants (standard, Gaussian, Mixture of Gaussian, Categorical), Transformer, Diffusion, VAEs, GANs, and VLD, demonstrated perfect adherence, recording 0% systematic violation rates for both the instantaneous power cap and the cumulative energy constraint. This outcome indicates that $\mathcal{L}$ and $\mathbf{P}$ provide enough information during training to make the generation adhere to physical viable designs, without the need of explicitly enforcing this constraints.

The observed systematic violations were isolated to two models: **Tabsyn** and **FlowMatching**.

- **Tabsyn** exhibited the highest violation rate for the instantaneous power cap at 7.69%±0.74. This significant breach is attributed to the model's inability to capture the requisite smoothness and temporal dependencies essential for physically viable LPs. Consequently, its outputs are characterized by high-frequency noise, which frequently pushes the instantaneous power beyond the established physical ceiling.

- **FlowMatching** demonstrated lower, but persistent, violations, recording a power cap breach rate of $2.45\% \pm 0.23$. Its cumulative energy violation was negligible ($0.02\% \pm 0.001$). These violations suggest that while the model generally maintains the overall profile, minor, transient noise artifacts are introduced during the generation process, leading to systematic failure to perfectly satisfy the strict per-timestep power boundaries.

The contrasting performance underscores the challenge of balancing generative flexibility with physical fidelity. Models achieving 0% violation are inherently more suitable for real-world ICF applications, as they minimize the need for external post-processing and correction, which could otherwise introduce undesirable artifacts into the laser pulse profile.

### F.2 QUALITATIVE ANALYSIS OF GENERATIVE MODEL FAILURE

Table 1 provides quantitative evidence of performance failure. To provide the requested actionable insight, we present a qualitative analysis by correlating the failure mechanisms of each generative paradigm to specific artifacts in the generated laser pulses (l′). This analysis helps determine why certain models fail to meet the stringent physical requirements of ICF design.

**Flow-Based and Domain-Agnostic Models (FlowMatching, Tabsyn, VLD)**  The high m Errors ($\sim 15\%$ to $26\%$) for these models indicate a fundamental failure to learn a proper laser pulse structure. These approaches fail to capture inductive bias necessary to model temporal sequences with sharp constraints.

**Structural Discontinuity and Boundary Violation.** For **FlowMatching** and **Tabsyn**, the generated pulses often exhibit unphysical power discontinuities and severe, high-amplitude oscillation.

These artifacts violate the fundamental requirement for a smooth, physically stable power curve, rendering the LPs non-viable for implosion.

**Lack of Critical Feature Fidelity.** **VLD** generates LPs that are physically viable (low Energy Conservation Error) but fail to capture finer, critical detail of the pulse. This lack of definition means the generated pulses are structurally similar to the ground truth but fail to accurately set the necessary conditions for correct LP implosion.

**Generative Adversarial Networks (GANs)**   GANs demonstrate a strong ability to produce LPs that are structurally realistic and adhere to physical constraints (low Energy Conservation Error). However, they suffer from a well-known adversarial flaw that compromises the pulse's physical utility. While the generated LPs look realistic, they contain unphysical high-frequency noise. This noise represents rapid, non-smooth power changes. These rapid changes are physically impossible for the laser system to deliver and would introduce hydrodynamic instabilities. Future work with GANs should incorporate an explicit regularization or post-processing denoising to filter this high-frequency noise.

**Variational Autoencoders (VAEs)**   The LPs generated by VAEs exhibit a slightly higher m Error (2.75%) compared to the best performing Diffusion/LSTM models. This can be directly attributable to the VAE's objective of regularizing the latent space. VAEs learn a smooth latent distribution, which forces the generated samples to be a statistically average of the training data. This process seems to over-smooth the pulse, particularly the sharp transitions. The generated LPs have overly rounded or damped features, failing to accurately represent the local slope changes required to maintain pressure balance. This mechanism could also explain the model's lower diversity, as the pulses might be overly constrained by the latent space prior.

# G   EXAMPLE PULSE SHAPE GENERATION

We present some of the pulses generated by the different approaches. Figure 7a for Diffusion, Figure 7b for Masked Autoregressive Flow, Figure 8a for LSTM, Figure 8b for Transformer, Figure 9a for GaussianAR, Figure 9b for MixtureOfGaussianAR, Figure 11a for VAE, Figure 11b for GAN.

Figure 10b for Tabsyn, and Figure 10a for VLD. WE can observe that both Tabsyn and VLD fail to design meaningful LPs.

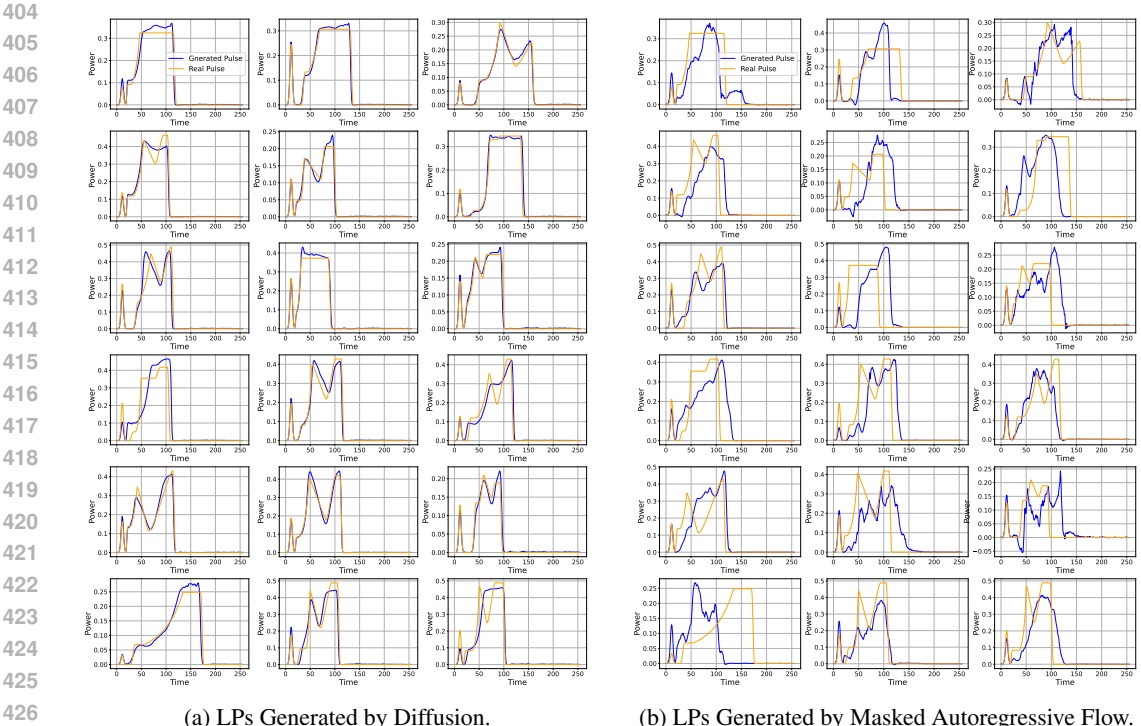

(a) LPs Generated by Diffusion.    (b) LPs Generated by Masked Autoregressive Flow.

Figure 7: LPs generated by Diffusion and FlowMatching

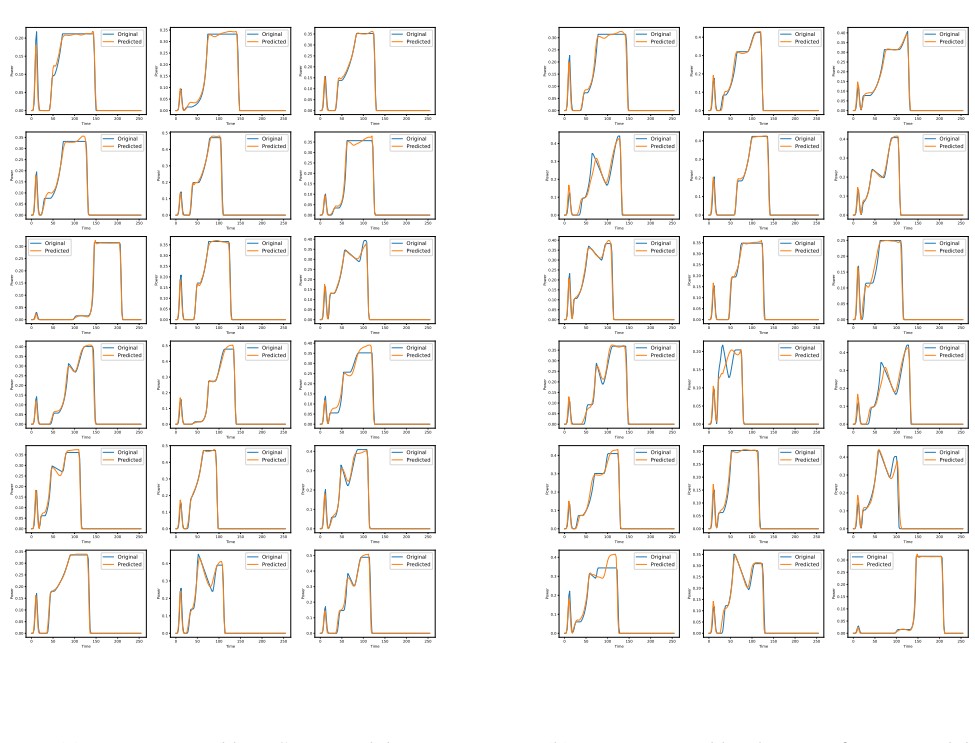

(a) LPs generated by LSTM model    (b) LPs generated by the Transformer model

Figure 8: LPs generated by predictive predictive auto-regressive models

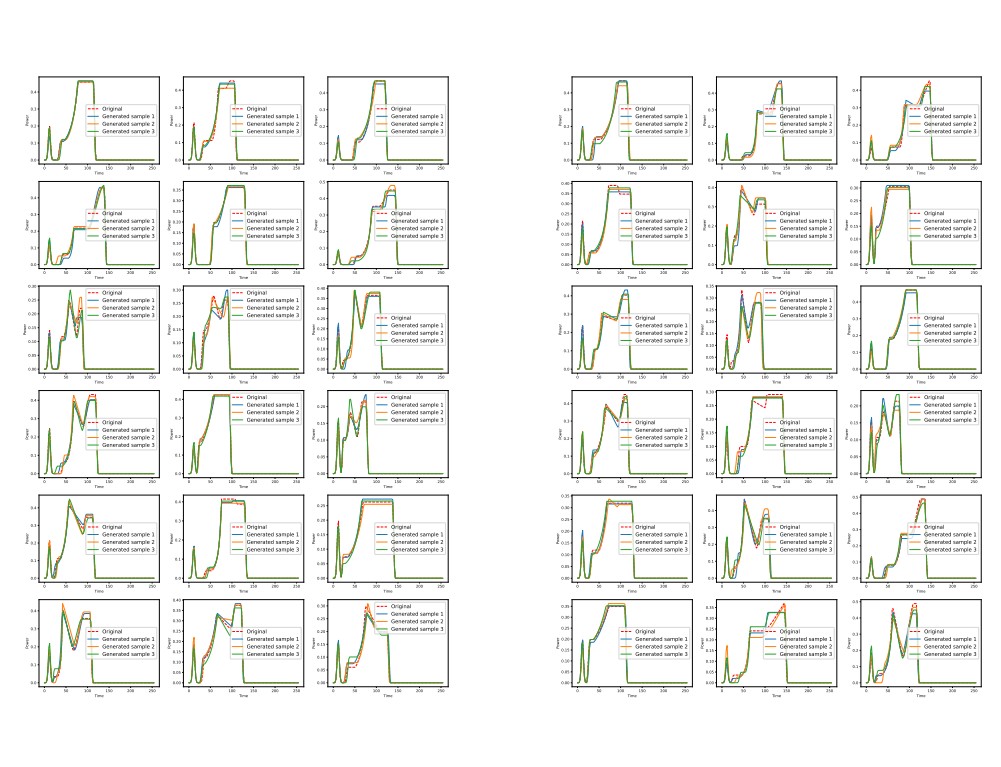

(a) LPs generated by GaussianAR model      (b) LPs generated by the MixtureofGaussianAR model

Figure 9: LPs generated by generative auto-regressive model

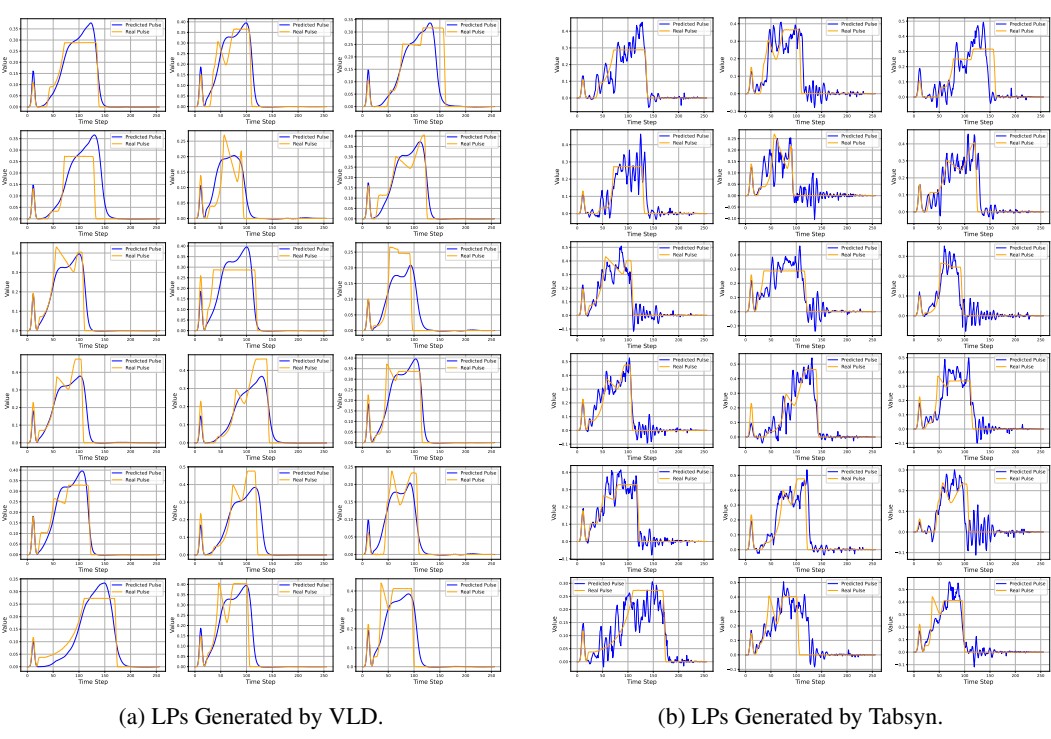

(a) LPs Generated by VLD.      (b) LPs Generated by Tabsyn.

Figure 10: LPs generated by VLD and TabSyn

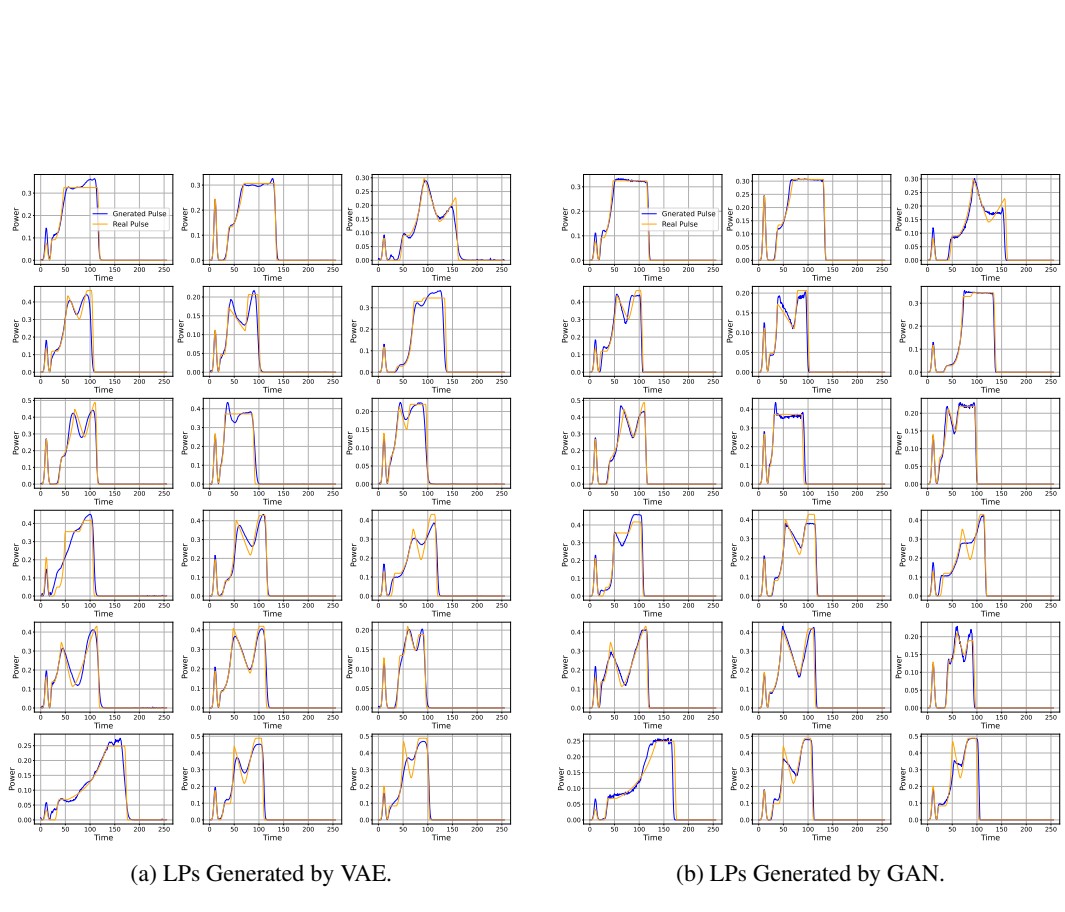

(a) LPs Generated by VAE.

(b) LPs Generated by GAN.

Figure 11: LPs generated by VAE and GAN

