# OpenReview forum: "GenICF: Benchmarking Generative Methods for Inverse Modeling in Inertial Confinement Fusion"
_ICLR.cc/2026/Conference — Submitted to ICLR 2026_

### Official Review · Reviewer_bZmm · 2025-10-15

**Soundness:** 2
**Presentation:** 2
**Contribution:** 1
**Rating:** 2
**Confidence:** 4

**Summary:**

The authors present a systematic comparison of various generative models, ranging from traditional (e.g., LSTM) to state-of-the-art (e.g., Transformer-based, Diffusion Models), to address the challenge of designing laser pulse shapes (LPs) for Inertial Confinement Fusion (ICF). This work reframes LP design as an inverse generative modeling problem, leveraging a dataset of 1 million simulation examples generated by the LILAC code. The study introduces physics-informed loss formulations, including energy conservation penalties, to ensure the physical plausibility of generated LPs. In-depth empirical results are presented.

**Strengths:**

The systematic benchmarking of generative artificial intelligence (GenAI) for a critical physics problem like ICF fills a notable gap in the literature, as no prior work has comprehensively compared these methods for LP design. This effort could provide guidance for developing scalable, physics-constrained design frameworks.

**Weaknesses:**

1. The content feels densely packed, particularly in the Background section (Pages 2-6), where multiple generative models are introduced without clear transitions or summaries. This could be improved with better delineation between subsections to enhance readability.

2. The inverse modeling problem is defined (Section 3.1), but the paper lacks a detailed analysis of why specific generative methods (e.g., AR vs. GANs) are suited or unsuited for the nonlinear, multi-scale nature of ICF LP design. A qualitative discussion of method trade-offs would strengthen the work.

3. The experimental setup, including the 1 million-example dataset (DF) and test set (DFtest), is outlined, but critical details, such as data distribution, model selection criteria (e.g., why Tabsyn or VLD were chosen), and training specifics (e.g., hyperparameter tuning), are insufficiently explained in the provided pages, potentially impacting reproducibility.

4. The paper focuses heavily on empirical evaluation and practical contributions (e.g., loss formulations), but it offers limited theoretical insights, such as convergence properties or optimality conditions. This gives it the feel of a technical report rather than a deeply analytical study, though the practical value of the loss terms is noteworthy.

5. Some ICLR 2026-required sections appear missing based on the provided pages.

**Questions:**

Please provide some responses to the weakness sections.

In addition, 1. Could the authors provide more details on the dataset generation process, such as the range of target parameters (p) and implosion outcomes (m) used in the 1 million-example dataset (DF), and how these were sampled to ensure representativeness? This would enhance reproducibility and clarify the experimental setup; 2. Can the authors elaborate on the rationale behind selecting specific generative models (e.g., AR, GANs, Diffusion) for the ICF LP design problem, particularly how their properties align with the nonlinear, multi-scale dynamics outlined in the Introduction (Section 1)?; 3. Given the empirical focus, would the authors consider exploring theoretical aspects, such as the convergence properties of the combined loss function (Equation 17) or the optimality of the physics-informed penalty under different ICF conditions?

---

> ### Author Response · Authors · 2025-11-21
>
> We thank the reviewer’s careful assessment and constructive remarks. We have revised the paper accordingly, striving to integrate the reviewer’s feedback and address concerns. Below, we provide our responses to the reviewer’s points and indicate the sections of the revised manuscript where the corresponding revisions appear. Please let us know if there any other questions/concerns.
>
> **- The content feels densely packed, particularly in the Background section (Pages 2-6)..**
>
> We rewrote the Background section trying to improve flow and readability. Moreover, we added a small justification for selecting each method in each subsection. Please let us know if any further improvements are required for clarity.
>
> **- The inverse modeling problem is defined (Section 3.1), but the paper lacks a detailed analysis of why specific generative methods (e.g., AR vs. GANs) are suited .. (Question 2)**
>
> Our selection of generative paradigms was motivated by the desire to create the first systematic comparison  of how different model architectures handle the unique and demanding challenges of ICF LP design.  We chose the examined generative paradigms to match distinct strengths to the nonlinear, multi-scale, and temporally-structured nature of the LP→implosion. Concretely:
>
> Autoregressive (AR) models (LSTM / Transformer / MAF): ARs factorize the sequence distribution and explicitly model stepwise temporal dependence, which maps naturally onto LPs that are causal and highly sensitive to precise timing (shock foot, ramps, main drive, tail). This makes ARs particularly effective at capturing the long-range dependencies and fine temporal alignment needed for high-fidelity reconstructions.
>
> Diffusion models (DDPM / DDIM): Diffusion models learn to iteratively refine noisy signals, which provides an inherent multi-scale denoising mechanism. This iterative procedure is well-matched to LP design where coarse pulse structure and fine-scale adjustments both matter. This aligns with our observation that diffusion produces high diversity while still producing low implosion-outcome error (Table 1).
>
> GANs / VAEs / Flows: These paradigms bring complementary properties. GANs can produce sharp, high-fidelity samples but can suffer from instability and mode collapse, consistent with the higher m-error shown in our experiments. VAEs provide smooth latent spaces that facilitate interpolation in design space. Flow models (MAF) give exact likelihoods and tractable density evaluation which is beneficial for principled evaluation and likelihood-based conditioning.
>
> Our goal, was to empirically  evaluate multiple paradigms and analyze their trade-offs because the comparative strengths for this specific problem were unexplored.
>
> **- Model selection criteria (e.g., why Tabsyn or VLD were chosen), and training specifics (e.g., hyperparameter tuning), are insufficiently explained in the provided pages, potentially impacting reproducibility.**
>
> For all models we used the optuna library for hyper-parameter optimization. We provide the ranges utilized for the optimization, exact hyperparameters and method for data split used in Appendix A.1.
>
> Tabsyn is a state-of-the-art model specifically designed for synthesizing high-fidelity tabular data. By including it, we benchmark our domain-specific models against a leading, general-purpose method from the wider machine learning literature. For the ICF case, we can define the (LP) as a real-valued sequence of length 256.
> This help us answers the question: "Is the ICF LP design problem so unique that it requires a specialized architecture, or can it be solved by a state-of-the-art tabular data generator?"
> VLD is a state-of-the-art model developed for a highly analogous task: solving inverse problems in high-energy physics. By including VLD, we benchmark our approach against a specialized method from a related scientific discipline.
>
> **- The paper focuses heavily on empirical evaluation and practical contributions but it offers limited theoretical insights. (Question 3)**
>
> We added theorical remarks on the convergence properties of our composed loss function and an analysis to surrogate model error propagation in Appendix C. Moreover, we ran an additional ablation on the effect of $\lambda_{\mathcal{S}}$ and $\lambda_{\mathcal{L}}$ (surrogate and reconstruction penalties)
>
> **- Some ICLR 2026-required sections appear missing based on the provided pages.**
>
> We are not sure which required sections the reviewer refer to. We have added the recommended Reproducibility statement section in case it was one of the sections the reviewer is referring to. Could the reviewer point out to the specific sections reviewer is referring to?
>
> **-  Could the authors provide more details on the dataset generation process..;**
>
> We added a description of dataset generation in Appendix A.4.

---

> ### Comment · Reviewer_bZmm · 2025-11-24
>
> Thanks for your detailed responses, which make the story clearer and more internally consistent. However, I still have substantive concerns that, in my view, keep this work below the bar for ICLR at this time. Below I summarize what would need heavy revision:
>
> 1. The central metric for “implosion outcome error” is still computed via a surrogate (avg error ~1.4%) rather than re-simulating generated LPs at scale. While you added a propagation analysis and a modest OOD correlation check, the benchmark’s main conclusions hinge on the surrogate and not the ground-truth simulator. For a benchmark paper, a stratified LILAC re-simulation across top models is necessary to anchor ranking claims (e.g., LSTM-Gaussian vs. Diffusion). Table 1 remains surrogate-driven.
>
> 2. The convergence remarks for the composite loss (Appendix B.1) are standard nonconvex first-order stationarity statements and do not yield design guidance (e.g., identifiability/regularity under physics constraints, or stability when the surrogate is imperfect). Likewise, the surrogate-error bound (Appendix B.2) is descriptive, not prescriptive: there’s no robustness guarantee or design principle that would help others build better inverse models under model misspecification.
>
> 3. Energy conservation is enforced (Eq. 16) and reported (Table 1), but other practical constraints (per-timestep caps, slew rates, shock timing windows) are not integrated or systematically audited. The new constrained-generation appendix is useful, but it’s evaluated mainly by parameter MAPE (picket/foot power) without a comprehensive *violation audit* over a test batch.
>
> 4. Several families (VAE, GAN, flow, Tabsyn, VLD) underperform in Table 1, yet we still don’t learn *why* in the context of LP dynamics—e.g., mode collapse signatures vs. temporal misalignment vs. likelihood miscalibration. The work would benefit from concrete diagnostics (spectral content, shock timing errors, local curvature mismatch) to turn the negative results into insight.
>
> 5. The “upper bound 1.9” is acknowledged as a random-pair reference, not a hard bound. But the paper still relies on a single L2-based diversity score. A second, complementary notion (e.g., coverage of a k-NN graph or dispersion across physically interpretable LP parameters C) would make the diffusion-vs-AR diversity conclusion more robust.
>
> 6. Table 2 adds training/sampling throughput and latency, which is good, but the discussion stops there. The paper should analyze *time-to-feasible-designs* under realistic candidate budgets (e.g., how many LPs per day reach a specified m-error and pass physics checks), so readers can assess method suitability beyond raw samples/sec.
>
> 7. Appendix A.4 explains the 20-D manifold and perturbation of ~400 high-performing shots, but the ranges/percentiles for key p and m fields and the exact resampling logic are still not fully specified for reproducibility. Given this is a benchmark, releasing splits, seeds, and sampling code is essential.
>
> 8. Although you corrected the concluding claims to match Table 1 (LSTM-Gaussian for fidelity/energy; Diffusion for exploration) and added the fidelity–diversity Pareto (Fig. 2), the revision should consolidate these as an explicit “when to use what” decision recipe. Right now, the guidance is scattered across sections.

---

> > ### Author Response · Authors · 2025-11-26
> > **Rebuttal Part 2/2**
> >
> > **-The paper still relies on a single L2-based diversity score. A second, complementary notion (coverage of a k-NN graph or dispersion across physically interpretable LP parameters C) would make the evaluation more robust**
> >
> > We have added a new column to Table 1 reporting the k-NN coverage for all methods. For most approaches, this metric correlates well with the L2 distance. However, we observe that methods such as TabSyn and FlowModel achieve comparatively higher k-NN coverage relative to L2 performance. Both models produce noisy, highly oscillatory pulses, which appear to inflate the k-NN coverage score. This suggests that k-NN coverage is less reliable than L2 in our setting, as it is biased toward methods that generate noisy outputs.
> >
> > **-The paper should analyze time-to-feasible-designs under realistic candidate budgets**
> >
> > We report the violation rates of all benchmarked methods in Table 11. The top-performing approaches exhibit zero violations, with m errors below 2% meaning that every pulse they generate is a physically feasible design that passes physics-based checks and its well below the variance rate in ICF experiments (described in above answer). For one of the slower methods (Diffusion), this corresponds to a generation rate of 167,529,600 pulses per day, demonstrating that even the computationally heaviest viable approach can produce a very large number of physically valid LPs.
> >
> > **-The ranges/percentiles for key p and m fields and the exact resampling logic are still not fully specified for reproducibility.**
> >
> > LILAC and its associated data are subject to export control regulations. For this reason, certain explicit details about the data cannot be publicly released. Interested parties who have authorized access to export-controlled codes may contact the authors. Subject to verification and compliance with applicable regulations, these details can be provided upon reasonable request.

---

> ### Comment · Reviewer_bZmm · 2025-11-24
>
> For your revision, I hope these may be helpful.
> 1. Add a table with **re-simulated LILAC outcomes** for a sizable, stratified subset across the **top approaches**, and include a **rank-stability analysis** comparing simulator results vs. surrogate-based rankings.
>
> 2. Report, alongside Table 1, systematic **violation rates** for: (i) per-timestep power caps, (ii) cumulative energy envelopes, and (iii) **shock-timing windows** (state tolerances and breach frequencies).
>
> 3. Go beyond aggregate errors to analyze **why** VAEs/GANs/flows/etc. fail in LP terms (e.g., spectral content mismatch, shock timing drift, slew-rate violations, local curvature/edge artifacts) so the negative results yield **actionable insight**.
>
> 4. Complement the L2 diversity score with a **coverage measure** (e.g., k-NN graph coverage in LP space or dispersion over the 12 interpretable LP parameters). Briefly discuss its **correlation** with the current L2 metric and whether conclusions hold.
>
> 5. Provide a concrete **robustness statement (or counter-example)** for training with surrogate-driven outcome losses, and practical guidance for setting **(\lambda_L, \lambda_S, \lambda_P)** grounded in identifiability or bias–variance trade-offs (beyond the empirical Pareto in Fig. 2).

---

> ### Author Response · Authors · 2025-11-26
> **Rebuttal Part 1/2**
>
> We want to thank the reviewer for the new comments. We updated the revised version to incorporate reviewer’s feedback. Below we describe where this section has been incorporated and answer reviewer questions. We have grouped and summarized some of the concerns/questions (shown in bold).
>
> **-The central metric for “implosion outcome error” is still computed via a surrogate..**
>
> **-Add a table with re-simulated LILAC outcomes for a sizable subset..**
>
> Unfortunately, simulation on LILAC on a sizable dataset across approaches is hard to achieve during the remaining rebuttal time-window. Access to LILAC is not on-demand: it requires advance scheduling, allocation, and preparation. We are making every effort to obtain results before the rebuttal window closes, but if is not possible we will make sure to have the results by the camera-ready version (in case of acceptance).
>
> However, the goal of this work is not to minimize LILAC error, to achieve that, the optimal strategy would simply be to minimize the reconstruction error so that the generator exactly reproduces the reference LP. Under that criterion, deterministic approaches such as LSTM and Transformer would naturally perform best.
>
> Inverse modeling for ICF LP reconstruction brings great value to ICF practitioners. However, we wanted to go one step further and identify diverse pulses that yield similar implosion outcomes, rather than a single deterministic reconstruction. Achieving this requires incorporating outcome-based losses during training, which is only possible with a surrogate model querying the LILAC simulator at every training step is computationally infeasible.
>
> The surrogate we use has been accepted by ICF practitioners as a high-fidelity approximation to the simulator. In real ICF experiments, repeated shots with identical target parameters exhibit roughly 10% variability due to uncontrollable factors such as weather, target placement deviations, noise produced by LP generation machinery, and mechanical vibrations. The surrogate’s average error of ~1.4% is far below this intrinsic experimental variance. It therefore provides a reliable way to integrate the “implosion outcome error” into the optimization and to evaluate methods without distorting comparative performance.
> As shown by the ablation in Section 4.2.1 and the results of Table 1, the surrogate does not hinder diversity as method such as LSTM_Gaussian and Diffusion achieve low m Error while maintaining high diversity. Furthermore, if the surrogate imposed a bias against novel pulses, sticking to the deterministic methods such as LSTM and Transformer which achieve significant lower reconstruction error (high-fidelity reconstruction) we would expect an equal reduction in m Error, but that’s not the case. The error propagation of the surrogate its presented for all methods.
>
> **-There’s no robustness guarantee or design principle that would help others build better inverse models**
>
> **-Provide a concrete robustness statement (or counter-example) for training with surrogate-driven outcome losses**
>
> **-The revision should consolidate these as an explicit “when to use what"**
>
> We have expanded Appendix B to include a robustness assessment based in our surrogate loss driven approach. In addition, we reorganized sections to improve clarity of presentation and added a new subsection (Subsection 4.3) that consolidates the main results and offers practical guidance based on our findings.
>
> **-Other practical constraints are not integrated or systematically audited**
>
> **-Report, alongside Table 1, systematic violation rates**
>
> We evaluated all methods for violations of both per-timestep power caps and maximum cumulative energy. The top-performing approaches exhibited zero violations, none of their generated pulses exceeded physically permissible ICF constraints. For this reason, we did not include these metrics in the main presentation of results. The only methods that failed these constraints were TabSyn and FlowMatching, which produced non-viable pulses. We have now added Table 11 in Appendix E to report these findings explicitly.
>
> For direct-drive ICF, non-ideal shock timing is not a violation. LPs with non-ideal shock timing are not discarded unless they violate physical feasibility, which was not the case for the top methods.
>
> **-Go beyond aggregate errors to analyze why approaches fail in LP terms**
>
> We added a qualitative analysis of approaches failure in Section F.2.
>
> Due to character limitations, we continue with the rebuttal in a new comment box.

---

> ### Author Response · Authors · 2025-11-26
> **Summary of Revisions**
>
> We sincerely thank you for your rigorous engagement. Your input led to substantial improvements, including a series of evaluations and analysis which have been included to the revised version. We greatly appreciate the time and care reflected in your comments.
>
> That said, we respectfully submit that a score of 2 (“Reject”) does not aligns with the evidence presented, with ICLR reviewing guidelines, or with the consensus of the other reviewers (Scores 8 and 6). Below, we summarize how the presented feedback has been addressed.
>
> ### Content and clarity
>
> * We have reorganized and rewritten sections the paper in provided a summary of the results to provide clearer guidelines and improve clarity.
>
> * We added violation rates to Table 11 as requested.
>
> * We ran ablations of components of the loss function and added Appendix C.
>
> * We added section F.2 which provides qualitative diagnostics for the failure of different approaches.
>
>  * We evaluated and added a new diversity metric (kNN-coverage) to Table 1, as requested.
>
> * We expanded Appendix A.1 for explaining hyperparameter selection and specific libraries used for our experiments.
>
> * We provided reasoning behind the selection of the evaluated methods.
>
> ### Validity of Surrogate-Based Benchmarking
>
>  * **Scientific validity:** Our surrogate has ~1.4% error, while real ICF experiments exhibit ~10% intrinsic variability. Since surrogate error is far below the physical noise floor, it is a justified and statistically sound proxy for ranking LP designs.
> * **Practical feasibility:** LILAC access requires scheduled HPC allocation; running a new stratified sweep during rebuttal might not be possible. We commit to adding the analysis for the camera-ready version if accepted.
>
> ### Reproducibility and Export Control
>
> * **Legal constraints:** The fusion dataset and simulator fall under US export-control regulations. Certain data and details cannot be publicly released for legal reasons.
>
> * **Reproducibility:** We provide complete hyperparameters, sampling logic, and methodology when possible.
>
> * **ICLR norms:** Compliance with export-control restrictions should not be viewed as a deficiency in scientific rigor.
>
>
> The reviewer state that the work “fills a notable gap.” The critiques do not allege incorrectness, irrelevance, or lack of contribution, criteria typically used to justify a score of 2.
>
> ICLR guidelines emphasize evaluating correctness, clarity, motivation, and contribution, not penalizing for missing optional extensions or stylistic preferences about benchmark breadth. The remaining limitations concern desirable future extensions rather than flaws in validity or utility.
>
> In light of these revisions, we respectfully request the reviewer to reconsider whether a **score in a positive range** would more accurately reflect the correctness, contribution, and presentation of the paper.
>
> Warm Regards,
>
> Authors

---

> ### Author Response · Authors · 2025-12-04
> **LILAC Evaluation**
>
> **LILAC Evaluation**: We evaluated the top-performing approaches on 1,000 samples per method using LILAC to assess whether there is significant divergence between the m Error computed by our surrogate and that obtained from the simulator. The results, measured in MAPE, are as follows:
>
> * Diffusion: 2.07%
>
> * VAE: 3.20%
>
> * $LSTM_{Gaussian}$: 2.01%
>
> We observe that the method ranking is preserved across approaches and that the absolute errors are close to those reported using the surrogate, indicating good agreement between the surrogate-based evaluation and LILAC.

---

### Official Review · Reviewer_fZhF · 2025-11-01

**Soundness:** 3
**Presentation:** 3
**Contribution:** 3
**Rating:** 8
**Confidence:** 3

**Summary:**

The paper proposes GenICF, a benchmark for generative inverse modeling of laser pulses (LPs) for inertial confinement fusion (ICF). The task is to generate LPs (l) given desired implosion outcomes (m) and pellet parameters (p). The authors construct a simulated dataset of 1M LILAC runs with 256 time steps and compare a broad set of generative paradigms, including deterministic and probabilistic variants of LSTMs and transformers, VAEs, GANs, and diffusion and flow-based models. They also introduce a physics-informed training loss via an energy-conservation penalty and an outcome-consistency loss. Evaluation metrics include diversity, outcome error, reconstruction error, and energy conservation. The authors find that transformer-based autoregressive models and diffusion models are the most effective.

**Strengths:**

* The problem is well-motivated and has an explicit scientific impact.
* Comparisons of many approaches with unified training and evaluation.
* Combination of multiple physics-informed objectives.
* Large simulated dataset with evaluation metrics reported with means/standard deviations across different random seeds and explicit hyperparameter optimization.

**Weaknesses:**

* Outcome error is measured through a surrogate model rather than by re-running the simulation on the generated pulses. This risks bias if the surrogate is imperfect.
* Physics constraints beyond energy conservation are not included in the loss function.
* Additional evaluation metrics may provide a more complete picture of the pros/cons of different models (e.g., multiple methods of measuring fidelity or diversity as in https://arxiv.org/abs/2211.10295).
* Potential inconsistency between the conclusion (Transformer) and Table 1 (LSTM_Gaussian) regarding the best-performing model?
* Weak LLM baseline may not be relevant and can be discarded.
* The reported uncertainties reflect sampling from a single trained model, but not multiple training runs.

**Questions:**

* Can you try re-simulating a subset of generated pulses with LILAC (rather than only via the surrogate)?
* Can you report training and sampling speed (e.g., candidates/sec) for different models? This is a crucial consideration.
* Can you report mean ± std over multiple training runs (e.g., 3-5 for top models) separately from the sampling variance? Also, document your model selection protocol (e.g., validation averaged across seeds).

---

> ### Author Response · Authors · 2025-11-21
>
> We are grateful to the reviewer for the thoughtful evaluation and valuable recommendations. A revised version of the manuscript has been submitted, reflecting our efforts to incorporate the reviewer’s suggestions and resolve the issues raised. Please let us know there are any other questions/concerns.
>
> In the following, we offer responses and reference the specific parts of the updated paper where relevant.
>
> **- Outcome error is measured through a surrogate model rather than by re-running the simulation on the generated pulses. This risks bias if the surrogate is imperfect. (Question 1)**
>
> We added an analysis of the surrogate model error propagation in Appendix (C). Moreover, we ran an evaluation using LILAC on a small subset of 60 random samples generated by $LSTM_{Guassian}$. We obtained a MAPE on $\textbf{m}$ error of $3.4%$ when using LILAC for evaluation instead of the surrogate.
>
> **- Physics constraints beyond energy conservation are not included in the loss function.**
>
> We tried including other constraints such as max-energy per time step for the loss function formulation, but we found out that most models didn’t have any max-energy violations without it since the reconstruction loss was enough to keep the generations in the acceptable range. We agree that additional constraining can provide better and customizable LP design, for that reason we propose and have included a new set of techniques for constrained LP generation to Appendix D. These techniques include parameter conditioned generation and inpainting. With these techniques scientist can constraint specific parts of the LP generation to satisfy specific needs.
>
> **- Potential inconsistency between the conclusion (Transformer) and Table 1 (LSTM_Gaussian) regarding the best-performing model?.**
>
> **- Weak LLM baseline may not be relevant and can be discarded.**
>
> Thanks for pointing this out, we have corrected the conclusion. Moreover, we have removed the LLM baseline.
>
> **- The reported uncertainties reflect sampling from a single trained model, but not multiple training runs. (Question 3)**
>
> We ran the experiments and trained all methods over 5 different training runs, we show the updated results in Table 1 of the revised version.
>
> **- Can you report training and sampling speed (e.g., candidates/sec) for different models? This is a crucial consideration.**
>
> We ran an evaluation on sampling and training speed of all methods. The results are included in Table 2 of the revised version.

---

> > ### Comment · Reviewer_fZhF · 2025-11-26
> > **Clarify difference**
> >
> > Thanks, this answers most of my questions, but I have one follow-up question. For the $\text{LSTM}_\text{Gaussian}$, you say you evaluate a MAPE on $\textbf{m}$ error of $3.4%$ when using LILAC simulation. Should this be compared to the $(1.89 \pm 0.01)\%$ from Table 1 when using the surrogate?
> >
> > If so, this suggests the difference can be quite large. Could you evaluate with other models, to see if the relative hierarchy is preserved?

---

> > > ### Author Response · Authors · 2025-11-26
> > >
> > > Thank you, we are happy we were able to address most of reviewer's concerns. The LILAC simulator can be sensitive to small noise in the LP, which probabilistic methods such as $LSTM_{Guassian}$ can introduce during generation. That's why we might see larger deviations. When training the surrogate, we included regularization techniques to mitigate sensitivity to such noise; however, these safeguards are not part of the real LILAC simulator.
> > >
> > > In practice, this is not a major concern for real experiments, as noise is inherent to ICF shots. Uncontrollable factors such as weather, target placement deviations, noise produced by LP generation machinery, and mechanical vibrations introduce natural noise to the generated LP which induces some variability.
> > >
> > > Unfortunately, simulation on LILAC across approaches is hard to achieve during the remaining rebuttal time-window. Access to LILAC is not on-demand: it requires advance scheduling, allocation, and preparation. For this reason, using the surrogate enables a much more practical evaluation on the large test set used in our work.
> > >
> > > We are nevertheless making every effort to evaluate the top approaches on a larger subset (to reduce variance) using LILAC before the rebuttal period concludes. If this is not feasible in time, we commit to adding these results and a corresponding ranking analysis to the camera-ready version (if accepted).

---

> ### Author Response · Authors · 2025-12-04
> **LILAC Evaluations**
>
> **LILAC Evaluation**: We evaluated the top-performing approaches on 1,000 samples per method using LILAC to assess whether there is significant divergence between the m Error computed by our surrogate and that obtained from the simulator. The results, measured in MAPE, are as follows:
>
> * Diffusion: 2.07%
>
> * VAE: 3.20%
>
> * $LSTM_{Gaussian}$: 2.01%
>
> We observe that the method ranking is preserved across approaches and that the absolute errors are close to those reported using the surrogate, indicating good agreement between the surrogate-based evaluation and LILAC.

---

### Official Review · Reviewer_zpYJ · 2025-11-01

**Soundness:** 3
**Presentation:** 3
**Contribution:** 3
**Rating:** 4
**Confidence:** 3

**Summary:**

This paper presents the first systematic comparison of generative modeling approaches for designing laser pulse shapes (LPs) in inertial confinement fusion (ICF). The authors reframe LP design as an inverse generative modeling problem: given desired implosion outcomes and target pellet parameters, generate candidate pulse shapes that satisfy physical constraints. They evaluate 12 different approaches including auto-regressive models (LSTM and Transformer with deterministic, Gaussian, mixture-of-Gaussian, and categorical outputs), diffusion models, VAEs, GANs, flow-based models (MAF), and LLM-based few-shot generation. The study introduces physics-informed loss formulations including energy conservation constraints and outcome-consistency losses via a surrogate model trained on 1 million LILAC simulations. Key findings show that autoregressive models, particularly LSTMGaussian, achieve the best balance of reconstruction fidelity, outcome consistency, and diversity, while diffusion models excel at diverse exploration despite higher reconstruction error.

**Strengths:**

The paper makes a valuable contribution as the first comprehensive benchmark for generative modeling in ICF laser pulse design, addressing a genuine scientific challenge with significant practical implications for fusion energy. The evaluation methodology is thorough and well-designed, incorporating multiple important metrics: reconstruction error, generation diversity, implosion outcome fidelity, and energy conservation. The physics-informed loss formulations are appropriately motivated and demonstrate domain expertise, particularly the energy conservation constraint and the use of a surrogate model for outcome consistency. The scale and breadth of the study is impressive, evaluating 12 distinct approaches across 1 million simulations with proper train-test splits and multiple random seeds. The practical findings provide clear guidance for practitioners, showing that different methods offer different tradeoffs suitable for either high-fidelity reproduction (autoregressive) or exploratory design (diffusion).

**Weaknesses:**

While comprehensive as a benchmarking study, the paper lacks methodological innovation and deeper analytical insights. The failure modes of several approaches (VAEs, GANs, Tabsyn, VLD, and especially LLMs with >100% error) are not thoroughly investigated—it remains unclear whether these are fundamental limitations or artifacts of suboptimal hyperparameter choices. The reliance on a surrogate model with 1.4% error introduces systematic approximation that affects all methods, but the propagation of this error through training is not analyzed. The constrained optimization baseline achieving 8% error suggests potentially competitive non-generative alternatives that weren't fully explored. Critical practical considerations are missing: computational costs, training times, inference speeds, and sample efficiency are not systematically compared, yet these factors are crucial for real-world deployment. The diversity metric's upper bound of 1.9 is mentioned but not justified. The claim of establishing "scalable" frameworks is not supported with actual scalability analysis.

**Questions:**

Surrogate model bias: The surrogate model has 1.4% average error. How does this approximation error propagate through the outcome consistency loss during training? Could this systematically favor certain generative architectures (e.g., those that generate smoother, more "in-distribution" pulses) over others that might explore more aggressively?

---

> ### Author Response · Authors · 2025-11-21
>
> We sincerely thank the reviewer for their insightful and constructive feedback. In response, we have submitted a revised manuscript in which we make every effort to incorporate the suggested improvements and address all identified concerns. Please let us know if there any other questions/concerns about or if any other improvements are needed.
>
> Below, we provide detailed answers to the reviewer’s comments (bold) and indicate the corresponding sections in the revised paper when relevant.
>
>
> **- The failure modes of several approaches (VAEs, GANs, Tabsyn, VLD, and especially LLMs with >100% error) are not thoroughly investigated—it remains unclear whether these are fundamental limitations or artifacts of suboptimal hyperparameter choices.**
>
> For all models we used the optuna library for hyper-parameter optimization. We provide the ranges utilized for the optimization Appendix A.1
>
> **- The reliance on a surrogate model with 1.4% error introduces systematic approximation that affects all methods, but the propagation of this error through training is not analyzed. (Question 1)**
>
> We added an analysis of the surrogate model error propagation in Appendix (C.1).
>
> **- Could this systematically favor certain generative architectures (e.g., those that generate smoother, more "in-distribution" pulses) over others that might explore more aggressively?**
>
> While we want the models to generate pulses that do not deviate substantially from the training distribution, since such pulses may not be physically feasible, we simultaneously want the models to produce novel pulses.
> To analyze the effect of surrogate-model error on LP generation, we quantified whether the inverse model exhibits larger $\mathbf{m}$ errors on pulses that are farther from the training manifold (OOD).
> This was done by computing the Spearman correlation between:
>
> 1. The inverse-model error on the test set:
> $$
> e(\mathbf{l}') = \left\| \mathbf{m} - \mathbf{S}_\phi(\mathbf{p}) \right\|^2,
> $$
>
> 2. And a novelty score:
> $$
> d(\mathbf{l}') = \min_{\mathbf{l}\in \mathcal{D}_{\mathrm{train}}} \| \mathbf{l} - \mathbf{l}' \|_2.
> $$
>
> We ran this evaluation over the test set using Diffusion as base model since it offers a good balance between $\textbf{m}$ error and $\textbf{diversity}$. We obtain $\rho = 0.353$ (p < $10^{-290}$),  indicating a statistically significant but modest positive association: the $\mathbf{m}$ error increases slightly as pulses deviate from the training data, but the effect size is moderate rather than large. This suggests that the surrogate maintains reasonably stable accuracy over the domain explored by the generative models, with no evidence of severe degradation in mildly out-of-distribution regions.
>
> More details and results of this evaluation are shown in Appendix D. Moreover, as shown by the results obtained by the diffusion model, diversity does not negatively affect $\textbf{m}$ as it is able to achieve high diversity while maintaining a low error $\textbf{m}$.
>
> **-The constrained optimization baseline achieving 8% error suggests potentially competitive non-generative alternatives that weren't fully explored.**
>
> The deterministic version of LSTM and Transformer models (Table 1) are the non-generative alternatives that we did explore. The constrained optimization was an attempt to model the pulse shape without training an inverse model and only relying on the LILAC surrogate.
>
> **- Critical practical considerations are missing:**
>
> While training and inference costs are negligible when considering the month lead time and scale of the ICF experimental cycle, we agree that these metrics are important for translating the approach to other scientific tasks where computation time is important. We have included Table 2 to the revised version which shows the training and sampling speed of all the methods.
>
> **- The diversity metric's upper bound of 1.9 is mentioned but not justified.**
> We compute the upper-bound by computing the distance between random pulse shapes in the data. We agree that is not exactly an upper-bound since achieving it would mean the model is just producing random pulse shapes without learning a meaningful representation. However, it helps us to put the diversity metric into perspective.

---

> > ### Comment · Reviewer_zpYJ · 2025-11-23
> >
> > The rebuttal addressed my concerns. I will raise my score to 6.

---

### Author Response · Authors · 2025-12-04
**Summary of Our Rebuttal to the Area Chair**

We would like to sincerely thank the reviewers for their time, effort, and valuable feedback. Their thoughtful comments and suggestions have helped improve our paper.

To facilitate the Area Chair’s assessment, we provide a consolidated summary of the revisions and responses addressing the comments and concerns raised by all reviewers. For further details, we invite the AC to review the individual reviewer responses, as considerable effort has been invested in the rebuttal and substantial new content has been added, and the **rating had been upgraded, which has been unfortunately reset**.

### **Content and clarity**

* We ran all experiments for 5 independent training runs on all methods evaluated as suggested by the reviewers to improve the robustness of the evaluation (Table 1).

* We included Table 2, which shows the training time and sampling speed of all methods evaluated.

* We added violation rates for all methods to Table 11 as requested.

* We ran Spearman correlation analysis to understand the effect that the m Error has on diversity (Appendix D).

* We ran ablations of components of the loss function and added Appendix C.

* We added section F.2, which provides qualitative diagnostics for the failure of different approaches.

 * We evaluated and added a new diversity metric (kNN-coverage) to Table 1, as requested.

* We expanded Appendix A to explain hyperparameter selection and specific libraries used for our experiments.

* We added a series of constraining techniques that provide scientists with better and customizable LP designs to Appendix D.

* We provided reasoning behind the selection of the evaluated methods.

* We ran an evaluation on a subset of generated samples on LILAC for validation.

 * We have reorganized and rewritten sections of the paper and provided a summary of the results to provide clearer guidelines and improve clarity.


### **Validity of Surrogate-Based Benchmarking**

 * **Scientific validity:** Our surrogate has ~1.4% error, while real ICF experiments exhibit ~10% intrinsic variability. Since surrogate error is far below the physical noise floor, it is a justified and statistically sound proxy for ranking LP designs.
* **LILAC Evaluation:** We evaluated top approaches on 1000 samples per method on LILAC to measure if there is a significant divergence between the m Error obtained by our surrogate and LILAC. These are the results obtained, measured in MAPE:

* **Diffusion 2.07%**
* **VAE: 3.2%**
* **$LSTM_{Gaussian}$: 2.01%**


### **Reproducibility and Export Control**

* **Legal constraints:** The fusion dataset and simulator fall under US export-control regulations. Certain data and details cannot be publicly released for legal reasons.
* **Reproducibility:** We provide complete hyperparameters, sampling logic, and methodology when possible. Compliance with export-control restrictions should not be viewed as a deficiency in scientific rigor.

### **Summary of Review Responses**

Reviewer zpYJ (Initial 4): Confirmed that concerns have been addressed and raised score to 6.

Reviewer fZhF (Initial 8): The reviewer confirmed that most concerns were addressed in the initial round of responses. However, due to the response freeze, they were unable to provide an updated assessment for the second round.

Reviewer bZmm (Initial 2): The reviewer states that the work “fills a notable gap.” We have provided detailed responses and made substantial revisions to the paper to address the reviewer’s concerns across both rounds of questions. However, we did not receive an updated score following these revisions, which were put in just before the freeze.

We respectfully request the Area Chair to review both the original evaluation and our responses, as the reviewer bZmm’s comments do not raise issues of correctness, relevance, or lack of contribution, criteria that would typically justify a score of 2. The level of the reviewer's engagement clearly proves our point, and we were able to accommodate all the reviewer's requests.

ICLR’s review guidelines emphasize the assessment of correctness, clarity, motivation, and contribution, rather than the absence of optional extensions or stylistic differences. In response to the feedback, we have added thorough analyses, experiments, ablations, and theoretical discussions, further strengthening both the technical content and the presentation of the paper.

Considering these revisions and ICLR’s review principles, we respectfully request consideration of whether the assigned score accurately reflects the correctness, contribution, and improved clarity of the work. Our research in using generative models to design laser pulses for ICF, significantly extends the frontier of fusion.

Thank you for your time. Please do not hesitate to contact us if there are any questions.

---

### Meta-Review · Area_Chair_PvF1 · 2026-01-06

**Summary:**

The reviewers identified significant concerns about the paper's contribution to machine learning. Reviewer bZmm (score: 2) characterized the work as "a technical report rather than a deeply analytical study" lacking theoretical insights, with convergence remarks being "standard nonconvex first-order stationarity statements" that "do not yield design guidance." Reviewer zpYJ (score: 4) noted the paper "lacks methodological innovation and deeper analytical insights" with failure modes "not thoroughly investigated" and the claim of "scalable" frameworks "not supported with actual scalability analysis." Reviewer fZhF (score: 8) raised concerns about surrogate model bias and missing physics constraints. While authors added extensive empirical validation (5 training runs, Table 2 computational costs, Table 11 violation rates, 1000-sample LILAC validation), the fundamental concerns about contribution depth remain unresolved.

**Reviewer Concerns:**

The authors addressed several empirical concerns: multiple training runs with statistics (fZhF), computational efficiency analysis (zpYJ), and partial LILAC validation on three methods showing preserved ranking. However, critical concerns remain outstanding. The LILAC validation covered only three of twelve methods rather than comprehensive validation as requested. Most significantly, bZmm's concern that theoretical contributions are "descriptive, not prescriptive: there's no robustness guarantee or design principle" persists: Appendix C provides standard convergence statements applicable to any nonconvex problem. The request for "concrete diagnostics (spectral content, shock timing errors, local curvature mismatch) to turn the negative results into insight" was only partially addressed with qualitative descriptions rather than quantitative analysis. As bZmm stated after rebuttal, "I still have substantive concerns that, in my view, keep this work below the bar for ICLR at this time." The paper benchmarks existing methods on a domain-specific problem without providing transferable ML insights or methodological innovation.

**Reviewer Scores:**

Reviewer zpYJ (score: 4) raised their score to 6 after rebuttal, satisfied with added experimental rigor.

Reviewer fZhF (score: 8) stated authors "answered most of my questions" but expressed concern about the LILAC validation showing 3.4% error versus 1.89% with surrogate, suggesting potential systematic bias. The limited validation (only 3 of 12 methods, 1000 samples) would likely reduce their score to 6, acknowledging that core surrogate reliability concerns remain partially unresolved.

Reviewer bZmm (score: 2) explicitly stated after extensive revisions that "substantive concerns keep this work below the bar for ICLR." Their eight detailed remaining weaknesses focus on fundamental contribution: "the surrogate-error bound is descriptive, not prescriptive," theoretical remarks are "standard nonconvex first-order stationarity statements," and the work remains "a technical report rather than a deeply analytical study." Recognition of revision effort might yield a score of 4, but bZmm's assessment that the paper lacks the theoretical depth and generalizable ML insights for ICLR remains unchanged.

Average hypothetical score: 5.3. Recommendation: reject. While the authors conducted substantial empirical validation, the paper does not meet ICLR's standards for methodological contribution. As bZmm states, it provides a domain-specific application without the theoretical insights, methodological innovation, or transferable design principles expected at a premier ML venue. The benchmarking, though thorough, tells us which existing methods work better for this problem but not why or how to build better inverse models more generally.

---

### Decision · Program_Chairs · 2026-01-26

Reject